# M$^5$HisDoc: A Large-scale Multi-style Chinese Historical Document Analysis Benchmark

**Yongxin Shi, Chongyu Liu, Dezhi Peng, Cheng Jian, Jiarong Huang, Lianwen Jin**[*]
South China University of Technology
yongxin_shi@foxmail.com, liuchongyu1996@gmail.com
pengdzscut@foxmail.com, eechengjian@mail.scut.edu.cn,
jiarong_huang@outlook.com, eelwjin@scut.edu.cn

## Abstract

Recognizing and organizing text in correct reading order plays a crucial role in historical document analysis and preservation. While existing methods have shown promising performance, they often struggle with challenges such as diverse layouts, low image quality, style variations, and distortions. This is primarily due to the lack of consideration for these issues in the current benchmarks, which hinders the development and evaluation of historical document analysis and recognition (HDAR) methods in complex real-world scenarios. To address this gap, this paper introduces a complex multi-style Chinese historical document analysis benchmark, named M$^5$HisDoc. The M$^5$ indicates five properties of style, ie., Multiple layouts, Multiple document types, Multiple calligraphy styles, Multiple backgrounds, and Multiple challenges. The M$^5$HisDoc dataset consists of two subsets, M$^5$HisDoc-R (Regular) and M$^5$HisDoc-H (Hard). The M$^5$HisDoc-R subset comprises 4,000 historical document images. To ensure high-quality annotations, we meticulously perform manual annotation and triple-checking. To replicate real-world conditions for historical document analysis applications, we incorporate image rotation, distortion, and resolution reduction into M$^5$HisDoc-R subset to form a new challenging subset named M$^5$HisDoc-H, which contains the same number of images as M$^5$HisDoc-R. The dataset exhibits diverse styles, significant scale variations, dense texts, and an extensive character set. We conduct benchmarking experiments on five tasks: text line detection, text line recognition, character detection, character recognition, and reading order prediction. We also conduct cross-validation with other benchmarks. Experimental results demonstrate that the M$^5$HisDoc dataset can offer new challenges and great opportunities for future research in this field, thereby providing deep insights into the solution for HDAR. The dataset is available at https://github.com/HCIILAB/M5HisDoc.

## 1 Introduction

Historical documents are invaluable carriers of human cultural heritage, containing important information about human history, culture, and literary arts. Therefore, the preservation and analysis of historical documents are considered an urgent research topic. As the digital version of historical documents is easier to disseminate and retrieve, designing a digitization system [1] of historical documents is an effective way to make use of historical documents without causing irreversible damage. With the rapid development of Optical Character Recognition (OCR) technologies, significant progress [2, 3, 4] has been made in understanding historical documents, and some benchmarks [5, 6, 7] have been established. However, real-world scenarios present various challenges such as diverse page layouts,

---

[*]Corresponding author.

37th Conference on Neural Information Processing Systems (NeurIPS 2023) Track on Datasets and Benchmarks.

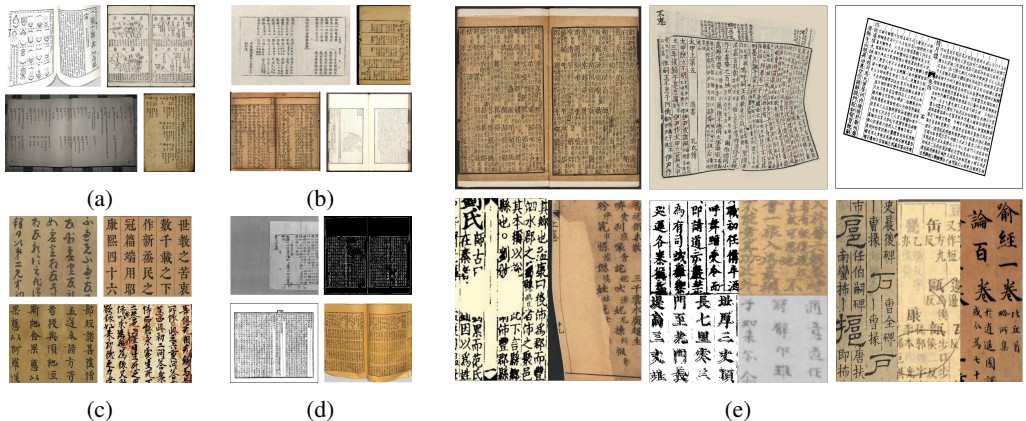

Figure 1: An overview of the proposed M⁵HisDoc dataset. For better visibility, please zoom in on the image. (a) Multiple layouts. (b) Multiple document types. (c) Multiple calligraphy styles. (d) Multiple backgrounds. (e) Multiple challenges, including dense texts, distortion, rotation, damage, image blurriness, and variations in font sizes.

poor image quality, multiple font styles, and severe distortion, which are rarely considered in the existing benchmarks. Although related methods have reported promising performance, they do not perform well on the aforementioned issues.

To fill this gap and advance the development of historical document analysis in real-world scenarios, we propose a complex multi-style Chinese historical document analysis benchmark, which contains 8,000 manually annotated images of historical documents. The characteristics of our dataset are as follows. (a) It features a wide range of styles, including multiple layouts (Fig. 1a), multiple document types (Fig. 1b), multiple calligraphy styles (Fig. 1c), multiple backgrounds (Fig. 1d), and multiple challenges (Fig. 1e). (b) The images in our dataset contain dense text, with an average of over 545 characters per image. (c) The dataset has a large character set of 16,151 categories, with zero-shot recognition scenario setting in the validation and test sets. (d) Both the annotations at the character-level and text-line-level are provided, including the text bounding box, text content, and the corresponding reading order. Therefore, M⁵HisDoc can be applied to a wide range of tasks, including text-line/character detection, recognition, and reading order prediction. (e) To better simulate real-world scenarios for historical document analysis, we perform random rotation, random distortion, and resolution transformations on the images of M⁵HisDoc-R, obtaining a more challenging subset, M⁵HisDoc-H.

Based on M⁵HisDoc, we benchmark several representative methods in the following tasks: text line detection, text line recognition, character detection, character recognition, and reading order prediction. We also conduct cross-validation with other benchmarks. The detailed experimental setup and results are presented in Sec. 4.

In conclusion, this paper presents two primary contributions. Firstly, we propose a complex multi-style Chinese historical document analysis benchmark, named M⁵HisDoc, which contains 8,000 images in multiple styles. The benchmark exhibits features of large scale, high text density, extensive character sets, and diverse styles. To the best of our knowledge, this is the first multi-style Chinese document analysis benchmark. Secondly, we conduct thorough baseline experiments using M⁵HisDoc, which demonstrate M⁵HisDoc presents novel challenges for historical document analysis. We strongly believe that M⁵HisDoc benchmark holds significant potential for future academic research endeavors.

## 2 Related works

### 2.1 Historical document benchmarks

The development of historical document analysis algorithms is closely linked to the progress of the corresponding benchmarks. ENP [8] contains 600 images of European historical newspapers, which are annotated with Unicode encoding of full text, layout information, type labels, and reading order.

DIVA-HisDB [9] provides 150 European medieval manuscripts labeled with text lines and layout information. HJDataset [10] is a large-scale dataset of Japanese historical documents, which contains more than 250,000 hierarchy layout elements of seven types. However, all of these datasets lack character-level annotations and do not specifically address Chinese historical documents. For Chinese historical documents, Xu et al. [7] established CASIA-AHCDB with more than 2.2 million text samples from 10,350 categories for character recognition in historical document images. MTHv2 [5] is a comprehensive dataset for Chinese historical documents, which can be used for text detection, text recognition, and reading order prediction. However, its document type is relatively homogeneous and contains only Buddhist scriptures. Similar to MTHv2, ICDAR 2019 HDRC-CHINESE [6] is a large structured dataset with only one document type, i.e., genealogy. Besides, SCUT-CAB [11] is a dataset of Chinese historical documents with complex layouts, but it only provides annotation for layout analysis without any information for text recognition.

## 2.2 Text detection

There are three main categories of text detection methods: regression-based, segmentation-based, and connected component-based. (a) Regression-based methods are inspired by object detection techniques such as Faster R-CNN [12] and SSD [13]. EAST [14] directly regressed text instances at pixel level and applied Non-Maximal Suppression (NMS) to obtain the final results. OBD [15] discretized blocks of unordered boxes and used Mask R-CNN [16] to address the inconsistent labeling issue of regression-based methods. MOST [17] proposed a text feature alignment module to dynamically adjust the receptive field and a position-aware NMS algorithm to further refine the detections. (b) Segmentation-based methods typically perform text detection by implicitly encoding text instances with pixel masks. PSENet [18] used a progressive scale-expansion algorithm on multi-scale segmentation maps to locate the text accurately. PAN [19] improved PSENet by a learning pixel aggregation algorithm to reduce computation cost. FCENet [20] predicted the Fourier signal vector of text instances and then reconstructed the text contour by Inverse Fourier Transform. DBNet++ [21] proposed an efficient adaptive scale fusion module based on DBNet [22] to enhance the accuracy of text detection. (c) Connected component-based methods typically detect individual text parts or characters first, followed by a post-processing procedure that links or groups them together to generate the final results. TextSnake [23] employed centerline and ordered disk to represent text instance. Zhang et al. [24] proposed a novel graph-based fusion network to adaptively fuse detection boxes to generate more accurate and holistic object instances for multi-oriented text detection. Besides, there are some text line detection methods specifically designed for historical documents. Mechi et al. [25] proposed an adaptive U-Net architecture for low-cost text line segmentation. Ma et al. [5] built upon Faster R-CNN [12] with two additional branches for character prediction and layout analysis, respectively. Then each individual detected character will be grouped into different text lines. Mechi et al. [4] used topological structural analysis to extract complete text lines.

## 2.3 Text recognition

Text recognition can be divided into three categories: CTC-based, attention-based, and segmentation-based. (a) CTC-based methods typically use the CTC algorithm to directly decode feature sequences into target sequences. CRNN [26] treated scene text recognition as a sequence recognition task. It employed CNN and RNN to extract text features, which are then decoded with the CTC algorithm. Ma et al. [5] improved CRNN by replacing the Bidirectional-LSTM with Residual-LSTM. ZCTRN [27] used a distance-based CTC decoder to match visual features with class embedding for zero-shot Chinese text recognition. SVTR [28] introduced local and global mixing blocks for extracting stroke-like features and inter-character dependence. (b) Attention-based methods use attention mechanisms to decode the target string sequence. In order to exploit the dependencies in both directions, the recognition decoder in ASTER [29] was extended into a bidirectional one. Sheng et al. proposed a self-attention-based sequence-to-sequence model NRTR [30] that does not rely on recurrence. To mitigate attention drift, Robust Scanner [31] incorporated a novel position enhancement branch and dynamically fused its outputs with those of the decoder attention module. Zhang et al. [32] proposed a cascade position attention mechanism to effectively handle the linguistic insensitive drift problem. (c) Segmentation-based methods first detect and recognize individual characters, then perform character combination with a post-processing algorithm, Liao et al. [33] and Peng et al. [34] are two typical methods. Liao et al. proposed a character attention FCN (Fully Convolutional Network) to predict the characters at pixel level, and then used a word formation module to group and arrange the pixels

to form the final word result. Considering that character-level annotation is insufficient, Peng et al. [34] proposed a weakly supervised learning paradigm, so that it can be trained with only transcript annotations.

## 2.4 Reading order prediction

The reading order prediction aims to arrange the text within a document in the correct order for reading. LayoutReader [35] is a sequence-to-sequence model using both textual and layout information, where it leverage the layout-aware language model LayoutLM [36] as encoder and modify the generation step in the encoder-decoder structure to generate the reading order sequence. Gu et al. [37] proposed a heuristic algorithm, Augmented XY Cut, to sort the input tokens. These methods primarily focus on modern documents. To the best of our knowledge, there is no work specializes for the reading order prediction of historical documents.

# 3 M$^5$HisDoc benchmark

## 3.1 Data collection

The objective of M$^5$HisDoc is to create an extensive benchmark for real-world scenarios in Chinese historical document analysis. To achieve this, we make significant efforts to gather a diverse collection of historical document images. Our data collection process consists of three main sources. Firstly, we carefully select 300 images from the training set of MTHv2 [5] and 700 images from SCUT-CAB [11], under the criteria that the selected images possess certain characteristics (such as dense text or complex layout). Secondly, we gather tens of thousands of scanned images from electronic ancient books available on the Internet (such as Harvard Library, and National Archives of Japan), encompassing 131 ancient books. From this collection, we manually curate 2,799 historical document images taken from some representative books. Thirdly, we conduct realistic photo shoots to simulate photographing situations. By selecting four physical Chinese ancient books, we capture 201 images using a scanner, considering various scanning angles and lighting conditions. In total, we obtain 4,000 images to establish the M$^5$HisDoc benchmark. The final dataset includes historical documents spanning different periods, such as the Han (206 BCE – 220 CE), Tang (618 – 907 CE), Song (960 – 1279 CE), Ming (1368 – 1644 CE), and Qing (1644 – 1912 CE) dynasties.

## 3.2 Data annotation

Initially, we train a character detector [12] and a character recognizer [38] using MTHv2 [5] and CASIA-AHCDB [7], respectively. We then utilize them for preliminary annotation. Afterward, we meticulously carry out the following procedures: (1) Adjust inaccurate character bounding boxes. (2) Proofread the character recognition results, which involve two rounds of checking. In the first round, two annotators proofread the character results on each image and point out the uncertain characters. In the second round, a domain expert finally correct the inconsistencies between the two annotation employees in the first round and the uncertain characters. (3) Determine which characters belong to the same text line. (4) Sort the text lines in the correct reading order. To ensure high-quality annotations, we implement a strict three-round check after the annotating process. The entire data collection, annotation, and check process takes approximately 4,000 person-hours.

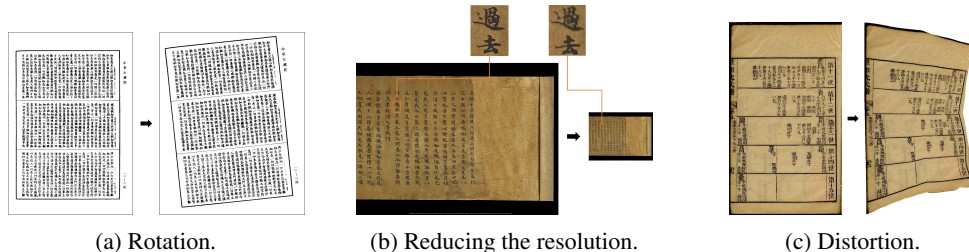

(a) Rotation.       (b) Reducing the resolution.       (c) Distortion.

Figure 2: Illustration of data processing to generate M$^5$HisDoc-H subset.

After the annotating is completed, we process the images in three ways to make the dataset more representative of some problems in real-world scenarios, such as image rotation, distortion, and blur. Firstly, we select 834 images with white backgrounds and randomly rotate them ± (5°∼15°). Secondly, we utilize DewarpNet [39] to randomly distort 50% of the images to simulate document distortion in the real world. Thirdly, We select 2,041 images featuring large character boxes, which are subsequently scaled down randomly, while maintaining a minimum edge size of 18 pixels. The processing scheme is shown in Fig. 2. We denote the pre-processed version and the post-processed version as **M⁵HisDoc-R** (Regular) and **M⁵HisDoc-H** (Hard), respectively. To ensure a fair evaluation, the two subsets are both divided into training, validation, and testing sets in a ratio of 2:1:1.

### 3.3 Dataset analysis

The comparisons of the statistics between M⁵HisDoc and other datasets are summarized in Table 1. We compare our dataset with previous datasets in three aspects, including text density, style types, and the number of character categories.

Table 1: Comparison with existing Chinese historical document datasets. Cls means The number of character categories. * indicates that we only consider the training set due to only the training set of ICDAR 2019 HDRC-CHINESE is available.

| Dataset | Images | Text lines/im | Char/im | Cls | Char box | Layout type | Doc type |
| --- | --- | --- | --- | --- | --- | --- | --- |
| MTHv1 [40] | 1,500 | 27.10 | 347.58 | 4,058 | ✓ | < 10 | 1 |
| MTHv2 [5] | 3,199 | 33.0 | 338.13 | 6,733 | ✓ | < 10 | 1 |
| IC19 HDRC [6]* | 11,715 | 35.64 | 211.95 | 8,353 | × | < 10 | 1 |
| CASIA-AHCDB [7] | - | - | - | 10,350 | - | - | 2 |
| M⁵HisDoc (Ours) | 8,000 | 50.48 | 545.92 | 16,151 | ✓ | > 20 | > 20 |

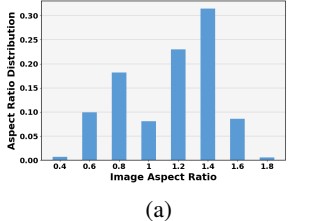

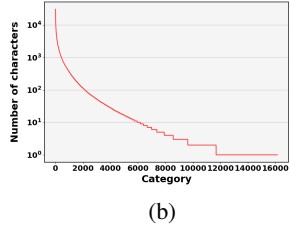

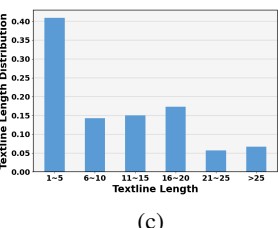

(a)            (b)            (c)

Figure 3: Statistics of M⁵HisDoc. (a) The aspect ratio of the images. (b) The number of characters per category. (c) The distributions of the text line length.

**Text density** As shown in Table 1, it is evident that the average number of characters and text lines in each image of M⁵HisDoc significantly exceeds that of existing datasets. The dense texts pose a significant challenge for both detection and recognition tasks.

**Style types** (a) Layout types: As shown in Fig. 1a, M⁵HisDoc dataset contains more than 20 types of layouts, which is more diverse than the existing benchmarks. The multiple types of layout pose a significant challenge for text detection and reading order understanding tasks. (b) Document types: M⁵HisDoc contains more than 20 document types, as are shown in Fig. 1b. In contrast, existing datasets typically have a single document type. For example, MTHv2 [5] and ICDAR 2019 HDRC-CHINESE [6] contain only Buddhist scriptures and genealogy, respectively. M⁵HisDoc, however, is sourced from a wide range of materials, including medical books, anthologies, ancient dictionaries, and so on. (c) M⁵HisDoc dataset encompasses a diverse array of Chinese calligraphic styles, including cursive, clerical, regular, and running scripts, which are illustrated in Fig. 1c. The significant variation in calligraphy styles poses a major challenge for the recognizer, requiring a robust system to accurately process them. (d) Background types: As shown in the Fig. 1d, M⁵HisDoc contains various background types. (e) M⁵HisDoc presents various challenges, including dense texts, distortion, rotation, damage, image blurriness, and variations in font sizes, as illustrated in Fig. 1e. Therefore, we can potentially avoid overfitting the historical document digitization system to

a specific type, ensuring that our dataset is more representative and comprehensive for real-world applications.

**The number of character categories** M$^5$HisDoc contains 16,151 categories, which is at least 1.6 times more than other existing datasets. The multitude of categories benefit a lot from the diverse range of our data sources, as documents of different types have their own unique character sets. In addition, as the character categories of historical document digitization systems often struggle to cover the ones in real-world applications, we set up zero-shot recognition scenarios in M$^5$HisDoc, where some character categories in the validation and test sets are not presented in the training set. The specific distribution of category are shown in Table 2, we can see that the validation set and the test set contain 1,853 and 1,851 categories that are not presented in the train set, respectively.

Table 2: Category distribution of M$^5$HisDoc.

| Dataset | Category | Zero shot category |
|---------|----------|--------------------|
| Training set | 12,779 | - |
| Validation set | 9883 | 1853 |
| Test set | 9748 | 1831 |
| All | 16,151 | - |

We also calculate the aspect ratio of the images, the number of characters per category, and the distributions of the text line length. As shown in Fig. 3a, the aspect ratio of the images varies significantly, ranging from less than 0.4 to more than 1.8. This is because M$^5$HisDoc contains a variety of styles and layouts. The number of samples per category exhibits a clear long-tail distribution, as demonstrated in Fig. 3b. The category with the largest number of samples consists of over 30,000 instances, whereas the category with the fewest has samples fewer than 3. It can also be observed from Fig. 3c that there exists a notable diversity in text length in M$^5$HisDoc. About 40% of texts exhibit a length of fewer than 6 characters, while about 7% of texts surpass the threshold of 25 characters. Some texts are extremely long, with the longest text containing 58 characters.

## 4 Experiments

### 4.1 Text line detection

**Setting** To provide a comprehensive evaluation of the text line detection on M$^5$HisDoc, we benchmark three categories of methods, including regression-based [16, 41, 15], segmentation-based [18, 19, 20, 21] and connected component-based [23] approaches. We use ResNet-50 [42] with a FPN pre-trained on ImageNet [43] as our backbone. All models are optimized using AdamW [44] with an initial learning rate of 1e-4. We train the models with a batch size of 8 for 160 epochs, and the learning rate decays by a factor of 0.1 at epochs 80 and 128. Various data augmentation techniques are applied, including random scale, color jitter, horizontal flip, random crop, and rotation. During training, the long sides of the training images are scaled to a fixed size of 1,333, while the short sides are randomly scaled to different scales (704, 736, 768, 800, 832, 864, 896). During testing, the long and short sides are resized to 1,333 and 800, respectively. The experimental evaluation metrics included precision, recall, and F1 score at different IoU thresholds (0.5, 0.6, 0.7). In Table 3, we also report the evaluation results of end-to-end performance using the normalized edit distance (1-NED) metric [45], by passing the detected text through the CRNN [26] model described in Sec. 4.2.

**Results and analysis** **(1) The extensive presence of lengthy and densely texts in M$^5$HisDoc poses challenges for segmentation-based methods.** As shown in Table 3, all methods can achieve considerable performance under the 0.5 IoU threshold. However, as the IoU threshold increases, segmentation-based models exhibit a substantial decline in text line detection. This because the presence of long texts introduces inaccuracies in the segmentation process, while dense texts make these methods prone to predicting fused results. **(2) The inclusion of rotated or distorted text in M$^5$HisDoc-H substantially diminishes the recall performance of regression-based methods.** The reason is that he proposals generated by the conventional Region Proposal Network (RPN) in regression-based methods remain horizontal and rectangular, which exhibit substantial overlap

with surrounding text instances. They will be easily suppressed by NMS and subsequently increase missing detection. **(3) High IoU detection is crucial for historical document analysis.** The last columns of Table 3 demonstrate that while the detection models show relatively good performance, the $1 - NED$ in the end-to-end systems falls short of expectations. This is largely due to the absence of accurate detection of texts, which leads to the omission of certain characters or the inclusion of surrounding characters.

Table 3: Results of text line detection, in the format of $M^5$HisDoc-H/$M^5$HisDoc-R test set.

| Type | Method | Venue | IoU thres | Precision↑ | Recall↑ | F1-score↑ | 1-NED↑ |
|---|---|---|---|---|---|---|---|
| Regression-based | Mask R-CNN [16] | CVPR'16 | 0.5 | 88.66/97.73 | 80.46/92.84 | 84.36/95.22 | |
| | | | 0.6 | 88.04/97.36 | 79.89/92.50 | 83.77/94.87 | 66.17/83.85 |
| | | | 0.7 | 86.61/96.44 | 78.60/91.62 | 82.41/93.97 | |
| | Cascade R-CNN [41] | CVPR'18 | 0.5 | 91.61/**98.41** | 82.37/92.88 | 86.74/**95.57** | |
| | | | 0.6 | 91.07/**98.10** | 81.89/92.59 | 86.24/95.26 | 69.68/**84.43** |
| | | | 0.7 | 89.94/97.35 | 80.86/91.88 | 85.16/94.54 | |
| | OBD [15] | IJCV'21 | 0.5 | 94.73/97.31 | 81.52/91.35 | 87.63/94.24 | |
| | | | 0.6 | 91.91/96.55 | 79.10/90.63 | 85.02/93.49 | 71.56/82.71 |
| | | | 0.7 | 85.99/94.66 | 74.00/88.85 | 79.55/91.66 | |
| Segmentation-based | PSENet [18] | CVPR'19 | 0.5 | 80.49/95.31 | 85.91/**94.51** | 83.11/94.90 | |
| | | | 0.6 | 76.14/91.03 | 81.26/90.26 | 78.61/90.65 | 63.77/80.60 |
| | | | 0.7 | 65.60/79.08 | 70.01/78.42 | 67.73/78.75 | |
| | PAN [19] | ICCV'19 | 0.5 | 92.83/93.59 | 80.48/88.58 | 86.22/91.02 | |
| | | | 0.6 | 88.70/86.91 | 76.91/82.27 | 82.39/84.53 | 68.91/75.10 |
| | | | 0.7 | 77.03/67.90 | 66.79/64.27 | 71.55/66.03 | |
| | FCENet [20] | CVPR'21 | 0.5 | 87.92/91.82 | 84.67/87.70 | 86.27/89.71 | |
| | | | 0.6 | 80.17/86.16 | 77.21/82.30 | 78.66/84.19 | 67.15/72.84 |
| | | | 0.7 | 66.80/75.67 | 64.33/72.28 | 65.55/73.94 | |
| | DBNet++ [21] | T-PAMI'22 | 0.5 | 91.81/89.78 | **89.76**/86.83 | 90.77/88.28 | |
| | | | 0.6 | 63.92/65.15 | 62.49/63.01 | 63.20/64.06 | 73.75/70.97 |
| | | | 0.7 | 39.01/38.10 | 38.14/36.84 | 38.57/37.46 | |
| Connected component-based | TextSnake [23] | ECCV'18 | 0.5 | **95.46**/93.25 | 86.67/88.74 | **90.85**/90.94 | |
| | | | 0.6 | 93.34/89.99 | 84.74/85.63 | 88.83/87.75 | **75.01**/74.76 |
| | | | 0.7 | 89.06/86.15 | 80.86/81.98 | 84.76/84.01 | |

## 4.2 Text line recognition

**Setting**  To assess the performance of text line recognition methods on $M^5$HisDoc, we select several representative methods from three categories: (1) CTC-based, including CRNN [26], Ma et al. [5], and ZCTRN [27]. (2) Attention-based, such as ASTER [29], NRTR [30] and Robust Scanner [31]. (3) Segmentation-based, such as Peng et al. [34]. The models are optimized by AdamW [44] with a base learning rate of 4e-4 and decay to 4e-6 using CosineAnnealing. The models are trained for 50 epochs with a batch size of 32. Thin Plate Spline transformation (TPS) algorithm is applied to each text line image for pre-processing. The height of each image is normalized to 96 while ensuring that the width is limited to a maximum of 1,536 pixels. Evaluation metrics are correct rate (CR) and accurate rate (AR) [34].

Table 4: Results of text line recognition, in the format of $M^5$HisDoc-H/$M^5$HisDoc-R test set. CR: correct rate, AR: accurate rate. * indicates the annotation of the character bounding box is used in training.

| Type | Method | Venue | CR↑ | AR↑ |
|---|---|---|---|---|
| CTC-based | CRNN [26] | T-PAMI'16 | 90.22/91.49 | 90.09/91.36 |
| | Ma et al. [5] | ICFHR'20 | **91.29**/92.29 | **90.83/92.10** |
| | ZCTRN [27] | ICDAR'21 | 87.19/88.79 | 86.96/88.50 |
| Attention-based | ASTER [29] | T-PAMI'18 | 86.95/87.98 | 86.63/87.70 |
| | NRTR [30] | ICDAR'19 | 84.50/85.43 | 82.03/83.02 |
| | Robust Scanner [31] | ECCV'20 | 88.55/90.99 | 88.33/90.78 |
| Segmentation-based | Peng et al. [34] | TMM'22 | 88.16/88.38 | 88.06/88.35 |
| | Peng et al. [34]* | TMM'22 | 89.97/**93.03** | 89.24/92.01 |

**Results and analysis**  **(1) The large number of character categories (16,151) and relatively long texts make attention-based methods perform poorly on this task.** These characteristics specifically hinder the effective modeling of contextual relationships by attention-based methods, and leading to significant attention drift issues[46]. Contrarily, CTC-based methods autonomously explore decoding paths and are less impacted by the aforementioned challenges [47], resulting in better performance. **(2) The style diversity presented in $M^5$HisDoc results in a significant performance gap between weakly supervised segmentation-based methods and fully supervised methods.** The existence of

various backgrounds and calligraphy styles introduces challenges for segmentation-based methods to achieve accurate character detection performance using weak supervision, a characteristic not commonly observed in other historical document datasets[5].

## 4.3 Character detection

**Setting** We evaluate the performance of a typical two-stage method, Faster R-CNN [12], and two one-stage methods, YOLOv3 [48] and YOLOX [49] on character detection. The models are optimized using SGD with an initial learning rate of 0.001. We train the models for 160 epochs with a batch size of 8, and the learning rate decays by a factor of 0.1 at the 128th and 144th epochs. In the training and testing phases, the longer and shorter sizes are adjusted to 1,333 and 800, respectively. The evaluation metrics of this task are the same as Sec. 4.1. Moreover, we crop the detected characters and pass them through the SwinTransformer-based [50] character recognizer in Sec. 4.4 to evaluate the end-to-end character recognition performance.

**Results and analysis** As depicted in Table 5, there is minimal disparity in performance between the one-stage and two-stage methods for character detection. **(1)** The performance gap between character-level detection schemes in M$^5$HisDoc-R and M$^5$HisDoc-H is significantly smaller than the disparity observed in text line detection as presented in Table 3. This can be attributed to the smaller scale variations of each character and the less sensitivity of character detection to rotation and distortion. **(2)** The character-based methods demonstrate comparable performance on both versions of M$^5$HisDoc in comparison to the results in Table 3.

In conclusion, the challenges posed by rotation, distortion, and other factors in M$^5$HisDoc-H significantly impact text-line-level approaches. Conversely, character-level approaches demonstrate relatively higher robustness in tackling these challenges.

Table 5: Results of character detection, in the format of M$^5$HisDoc-H/M$^5$HisDoc-R test set. "Top-1 acc" is the metric of extracting the detected bounding box and feeding it into the SwinTransformer-based character recognizer. * indicates the accuracy of the missed and overchecked characters is set to 0, otherwise only the characters matching on GT are considered.

| Type | Method | Venue | IoU thres | Precision↑ | Recall↑ | F1-score↑ | Top-1 acc↑ | Top-1 acc↑* |
|------|--------|-------|-----------|-----------|---------|-----------|-----------|-------------|
| Two-stage | Faster R-CNN [12] | NIPS'15 | 0.5 | 96.36/97.75 | **98.46**/98.53 | **97.40**/98.14 | | |
| | | | 0.6 | 95.27/96.91 | **97.34**/97.69 | 96.29/97.30 | 93.01/94.92 | |
| | | | 0.7 | 91.84/94.46 | **93.83**/95.21 | 92.82/94.83 | | 88.29/91.46 |
| One-stage | YOLOv3 [48] | arXiv'18 | 0.5 | **99.20**/**99.21** | 94.71/95.88 | 96.90/97.52 | | |
| | | | 0.6 | **98.61**/**98.77** | 94.14/95.45 | **96.32**/97.08 | **93.69**/**95.28** | 88.04/90.67 |
| | | | 0.7 | **96.43**/**97.19** | 92.06/93.93 | **94.20**/95.53 | | |
| | YOLOX [49] | CVPR'21 | 0.5 | 96.66/97.14 | 98.04/**99.17** | 97.35/**98.15** | | |
| | | | 0.6 | 93.77/96.48 | 95.11/**98.49** | 94.44/**97.47** | 93.57/95.06 | **88.73**/**91.60** |
| | | | 0.7 | 84.16/94.59 | 85.36/**96.56** | 84.76/**95.56** | | |

## 4.4 Character recognition

**Setting** The character recognition model involves utilizing a backbone network, followed by a fully connected layer for final classification. In this study, we analyze the influence of various backbones on the accuracy of the recognition results, including CNN-based [42, 51, 52] and Vision-Transformer-based backbones[53, 50]. The models are optimized by AdamW [44] with a base learning rate of 0.001 and decay to 1e-6 using CosineAnnealing. We train the models with a batch size of 1,024 for 90 epochs. The learning rate for the first 5 epochs warmup from 1e-4 to 1e-3. All models are trained using RandAugment [54]. The input images are all scaled to $96 \times 96$. Evaluation metrics are based on top-1 accuracy, top-5 accuracy, and **macro accuracy**[55] that refers to calculating the accuracy for each category individually and then averaging them.

**Results and analysis** As shown in Table 6, all models demonstrate considerable top-1 accuracy. However, there are still two prominent challenges in this task.**(1) Character recognition with similar morphology.** All models achieve high top-5 accuracy, indicating their ability to effectively limit the recognition results within a range of five potential candidates. Nonetheless, the Top-1 accuracy exhibits a significant decline due to errors in discerning similar Chinese characters. **(2) Long-tail distribution.** The macro accuracy of all models is noticeably lower than the top-1 accuracy. This

observation underscores the presence of the long-tail distribution problem, where characters with limited training samples cannot be accurately recognized.

In real-world historical document analysis scenarios, numerous similar morphology and uncommon characters exist, which pose challenges to character recognition. Accurately recognizing these characters becomes crucial in addressing the aforementioned issues. Compared to existing benchmarks, our research incorporates a wider range of visually similar and uncommon character categories, expanding the class number in $M^5$HisDoc, thereby providing a more comprehensive representation of these challenges.

Table 6: Results of character recognition, in the format of $M^5$HisDoc-H/$M^5$HisDoc-R test set.

| Type | Method | Venue | Top-1 acc↑ | Top-5 acc↑ | Macro acc↑ |
|---|---|---|---|---|---|
| CNNs | ResNet50 [42] | CVPR'16 | 94.40/95.07 | 98.18/98.46 | 69.61/73.96 |
| | RegNet [51] | CVPR'20 | 94.56/95.17 | 98.30/98.48 | 69.23/72.94 |
| | ConvNeXt [52] | CVPR'22 | 94.61/95.19 | 98.28/98.52 | **71.14/75.06** |
| Vision Transformer | ViT [53] | ICLR'21 | 94.09/94.63 | 98.28/98.21 | 65.27/68.10 |
| | SwinTransformer [50] | ICCV'21 | **94.68/95.22** | **98.33/98.56** | 71.08/74.34 |

## 4.5 Reading order prediction

**Setting** We conduct experiments on $M^5$HisDoc for reading order prediction with both rule-based and learning-based methods. The rule-based methods include Augmented XY Cut [37] specifically designed for layout analysis in modern documents, and a heuristic algorithm by ourselves. This algorithm arranges the center coordinates of text lines in a left-to-right, top-to-bottom order (The detailed procedure is presented in the Appendix), which is designed mainly based on the fact that Chinese historical documents are typically read in such a manner. For the learning-based method, we employ LayoutReader [35] for comparison. As all of the aforementioned methods exclusively take horizontal rectangle bounding boxes as input, we utilize horizontal minimum enclosing rectangle boxes of each text as inputs. We utilize Average Relative Distance [35] (ARD) to evaluate the performance.

Table 7: Results of reading order prediction, in the format of $M^5$HisDoc-H/$M^5$HisDoc-R test set. ARD: Average Relative Distance.

| Type | Method | Venue | ARD↓ |
|---|---|---|---|
| Rule-based | Heuristic method (Ours) | - | **5.27/5.20** |
| | Augmented XY Cut [37] | CVPR'22 | 14.63/12.60 |
| Learning-based | LayoutReader [35](layout only) | EMNLP'21 | 8.68/5.49 |

**Results and analysis** **(1) The reading order prediction methods designed for modern documents face challenges in generalizing to the analysis of historical documents.** The results in Table 7 demonstrate a substantial disparity in performance between augmented XY Cut and the heuristic-based algorithm. This discrepancy can be primarily attributed to the fact that augmented XY Cut is tailored specifically for modern documents. However, due to the significant gap between historical and modern documents, this method struggles to generalize effectively to historical document analysis. In contrast, the heuristic algorithm, which leverages the distinctive characteristics of Chinese historical documents, exhibits notable advantages in predicting the reading order. **(2) $M^5$HisDoc-H has presented greater challenges to existing methods for reading order prediction.** We can also observe that augmented XY Cut and LayoutReader experience a notable decline in performance on $M^5$HisDoc-H. This is because the polygonal boxes of text in $M^5$HisDoc-H are more distorted compared to those in $M^5$HisDoc-R, resulting in substantial differences between the horizontal minimum enclosing rectangles and the original polygons. Hence, $M^5$HisDoc-H presents numerous cases of overlap with external horizontal rectangles, which pose challenges for these methods to model the relationships between each text line.

## 4.6 Cross-validation with other benchmarks

**Setting** We train our models on MTHv2 [5] using Cascade R-CNN [41], CRNN [26], YOLOX [49], and ConvNeXt [52] architectures, with the settings described in Sec. 4.1, 4.2, 4.3, and 4.4, respectively.

We train Cascade R-CNN and CRNN on IC19 HDRC [6], ConvNeXt on CASIA-AHCDB [7]. Subsequently, we perform cross-validation using the models trained on M$^5$HisDoc-H.

Table 8: Cross-validation between the models trained on other datasets and M$^5$HisDoc-H. The evaluation metric for text line/character detection is F-score at 0.5 IoU, the metrics for text line/character recognition are AR and top-1 accuracy, respectively.

| Tasks | Text line det (F1-score) | Text line reg (AR) | Char det (F1-score) | Char reg (top-1 acc) |
|---|---|---|---|---|
| MTHv2 → M$^5$HisDoc-H | 53.88 | 53.95 | 85.77 | 73.71 |
| M$^5$HisDocH → MTHv2 | **97.46** | **95.83** | **99.20** | **96.35** |
| IC19 HDRC → M$^5$HisDoc-H | 39.92 | 56.53 | - | - |
| M$^5$HisDocH → IC19 HDRC | **70.56** | **85.60** | - | - |
| CASIA-AHCDB → M$^5$HisDoc-H | - | - | - | 71.91 |
| M$^5$HisDocH → CASIA-AHCDB | - | - | - | **92.55** |

**Results and analysis**  The results presented in Table 8 demonstrate the outstanding performance of the model trained on M$^5$HisDoc-H when evaluated on MTHv2, IC19 HDRC and CASIA-AHCDB. Conversely, the model trained on other datasets faces difficulties in generalizing to M$^5$HisDoc-H. These are mainly due to the multiple sytles of M$^5$HisDoc and strongly indicate that M$^5$HisDoc-H introduces novel challenges to the field, and also reflect its greater representativeness.

## 5   Limitation

Although significant efforts have been made to capture a diverse range of images from various perspectives and simulate realistic distortions through data synthesis, it is crucial to acknowledge that certain instances of distortion still remain uncovered in M$^5$HisDoc. This serves as a part of future work.

## 6   Conclusion

In this paper, we propose M$^5$HisDoc dataset, which is particularly notable for its diverse range of styles, including multiple layouts, document types, calligraphy styles, backgrounds, and challenges. Furthermore, we conduct an extensive benchmark evaluation of M$^5$HisDoc and perform a thorough analysis of the results. Our findings demonstrate that M$^5$HisDoc offers a more accurate representation of practical challenges encountered in historical document analysis, compared with existing benchmarks. Consequently, we firmly believe that this comprehensive dataset will provide researchers and developers with a valuable resource to advance the field of historical document analysis.

## Acknowledgments

This research is supported in part by NSFC (Grant No.: 61936003) and Science and Technology Foundation of Guangzhou Huangpu Development District (Grant No.: 2020GH17).

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
