# M$^5$HisDoc: A Large-scale Multi-style Chinese Historical Document Analysis Benchmark Supplementary Material

**Yongxin Shi, Chongyu Liu, Dezhi Peng, Cheng Jian, Jiarong Huang, Lianwen Jin**[*]
South China University of Technology
yongxin_shi@foxmail.com, liuchongyu1996@gmail.com
pengdzscut@foxmail.com, eechengjian@mail.scut.edu.cn,
jiarong_huang@outlook.com, eelwjin@scut.edu.cn

## 1 Datasheets for M$^5$HisDoc

### 1.1 Motivation

**For what purpose was the dataset created?**
The purpose of creating M$^5$HisDoc dataset is to advance the development of historical document analysis in real-world scenarios. In the field of historical document analysis, some benchmarks [1, 2, 3] have been established and some methods [4, 5, 6] have reported promising performance. However, these methods fail to adequately address the challenges posed by real-world scenarios, including diverse page layouts, poor image quality, multiple font styles, and severe distortion, which are rarely considered in the existing benchmarks. To fill this gap, we introduce M$^5$HisDoc, a comprehensive and intricate benchmark for Chinese historical document analysis. M$^5$HisDoc encompasses a wide range of document styles, including diverse layouts, document types, calligraphy styles, backgrounds, and associated challenges. In contrast to existing benchmarks, M$^5$HisDoc offers a more thorough representation of the aforementioned issues. Consequently, we firmly believe that this complex dataset will significantly promote the development of historical document analysis in real-world scenarios.

**Who created the dataset (e.g., which team, research group) and on behalf of which entity (e.g., company, institution, organization)?**
The M$^5$HisDoc dataset is created by the Deep Learning and Vision Computing Lab (DLVC-Lab) of South China University of Technology.

### 1.2 Composition

**What do the instances that comprise the dataset represent (e.g., documents, photos, people, countries)?**
The M$^5$HisDoc dataset comprises 8,000 images, accompanied by their respective annotation files. These images represent scanned or photographed historical documents and are stored in the Joint Photographic Experts Group (JPEG) format. The annotations, are stored in plain text (TXT) format, including bounding boxes for text lines/characters within the images, along with the corresponding text content. Furthermore, the texts are arranged in the correct reading order.

**How many instances are there in total (of each type, if appropriate)?**
The M$^5$HisDoc dataset comprises a collection of 8,000 images, with 403,824 text lines and 4,367,361 characters in 16,151 categories.

---

[*]Corresponding author.

37th Conference on Neural Information Processing Systems (NeurIPS 2023) Track on Datasets and Benchmarks.

**Does the dataset contain all possible instances or is it a sample (not necessarily random) of instances from a larger set?**
The $M^5$HisDoc dataset contains all possible instances.

**What data does each instance consist of?**
Each instance in the $M^5$HisDoc consists of an image along with corresponding annotations. These annotations include bounding boxes for text lines/characters within the images, along with the text content and the reading order between the texts.

**Is there a label or target associated with each instance?**
Yes. The label contains bounding boxes for text lines/characters, text content, and the reading order between the texts.

**Is any information missing from individual instances?**
NO. There is no missing information from individual instances in the $M^5$HisDoc dataset.

**Are relationships between individual instances made explicit (e.g., users' movie ratings, social network links)?**
There is no relationship between individual instances.

**Are there recommended data splits (e.g., training, development/validation, testing)?**
The two subsets of $M^5$HisDoc ($M^5$HisDoc-R and $M^5$HisDoc-H) are both divided into training, validation, and testing sets in a ratio of 2:1:1. The specific data splits can be found at `https://github.com/HCIILAB/M5HisDoc`.

**Is the dataset self-contained, or does it link to or otherwise rely on external resources (e.g., websites, tweets, other datasets)?**
The $M^5$HisDoc dataset is self-contained.

**Does the dataset contain data that might be considered confidential (e.g., data that is protected by legal privilege or by doctor– patient confidentiality, data that includes the content of individuals' non-public communications)?**
The $M^5$HisDoc dataset comprises historical document images along with their corresponding manual annotations and does not include any confidential data.

**Does the dataset contain data that, if viewed directly, might be offensive, insulting, threatening, or might otherwise cause anxiety?**
The dataset does not include any data that could be considered offensive, insulting, threatening, or potentially causing anxiety.

## 1.3  Collection Process

**How was the data associated with each instance acquired?**
The collection process is described in Sec. 3.1 of the main paper. The images we obtained from the Internet are sourced from some open-copyright (under CC license) websites, such as Harvard-Yenching Library (`https://library.harvard.edu/`), National Archives of Japan (`https://www.digital.archives.go.jp/`).

**What mechanisms or procedures were used to collect the data (e.g., hardware apparatuses or sensors, manual human curation, software programs, software APIs)?**
The annotation process is described in Sec. 3.2 of the main paper. And we specifically developed a web-based platform for data annotation, as illustrated in Fig. 1. With this platform, annotators are able to label text boxes, text content, and correct their reading order.

**If the dataset is a sample from a larger set, what was the sampling strategy (e.g., deterministic, probabilistic with specific sampling probabilities)?**
The sampling strategy employed in this study involved a manual selection process, where representative samples were carefully chosen from the larger set.

**Over what timeframe was the data collected?**
The data collection process spanned approximately 5 months.

**Did you collect the data from the individuals in question directly, or obtain it via third parties or other sources (e.g., websites)?**
Our data collection process consists of three main sources. Firstly, we carefully select 300 images

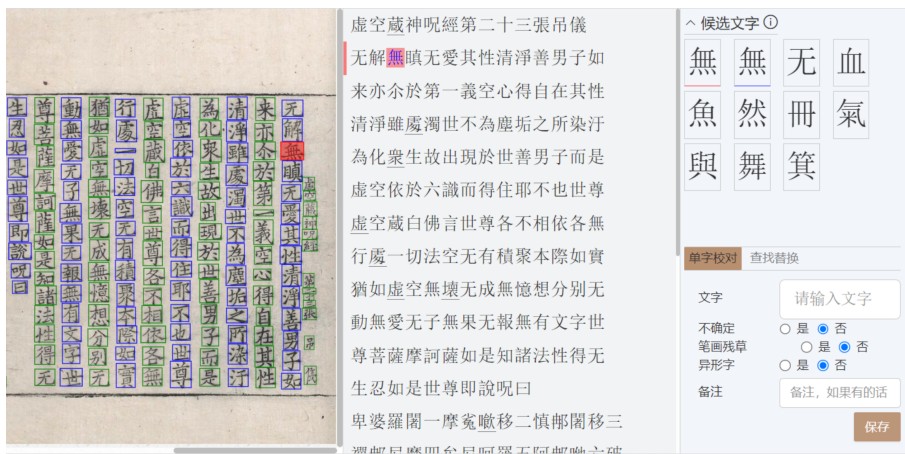

Figure 1: The web-based platform designed for data annotation.

from the training set of MTHv2 [1] and 700 images from SCUT-CAB [7]. Secondly, we gather tens of thousands of scanned images from electronic ancient books available on the Internet and manually curate 2,799 historical document images taken from some representative books. Thirdly, we conduct realistic photo shoots to simulate photographing situations. By selecting four physical Chinese ancient books, we capture 201 images using a scanner, considering various scanning angles and lighting conditions.

## 1.4 Preprocessing/cleaning/labeling

**Was any preprocessing/cleaning/labeling of the data done (e.g., discretization or bucketing, tokenization, part-of-speech tagging, SIFT feature extraction, removal of instances, processing of missing values)?**
To replicate real-world conditions for historical document analysis applications, we incorporate image rotation, distortion, and resolution reduction into $M^5$HisDoc-R subset to form a new challenging subset named $M^5$HisDoc-H.

**Was the "raw" data saved in addition to the preprocessed/cleaned/labeled data (e.g., to support unanticipated future uses)?**
Yes, the "raw" data for $M^5$HisDoc-H corresponds to $M^5$HisDoc-R.

## 1.5 Uses

**Has the dataset been used for any tasks already?**
No.

**What (other) tasks could the dataset be used for?**
The $M^5$HisDoc dataset can be used for various tasks of Chinese historical document analysis, including text line/character detection, recognition, and reading order prediction.

## 1.6 Distribution

**How will the dataset will be distributed (e.g., tarball on website, API, GitHub)?**
The $M^5$HisDoc dataset is available at https://github.com/HCIILAB/M5HisDoc.

**Have any third parties imposed IP-based or other restrictions on the data associated with the instances?**
No.

**Do any export controls or other regulatory restrictions apply to the dataset or to individual instances?**
All authors bear all responsibility for the $M^5$HisDoc dataset in case of violation of rights, etc. The

M$^5$HisDoc dataset should be used under Creative Attribution-NonCommercial-NoDerivatives 4.0 International (CC BY-NC-ND 4.0) License for non-commercial research purposes.

### 1.7 Maintenance

**Who will be supporting/hosting/maintaining the dataset?**
The dataset will be maintained by the Deep Learning and Vision Computing Lab (DLVC-Lab) of South China University of Technology.

**How can the owner/curator/manager of the dataset be contacted (e.g., email address)?**
Contact can be made via email at eelwjin@scut.edu.cn.

**Will the dataset be updated (e.g., to correct labeling errors, add new instances, delete instances)?**
If substantial errors are raised by dataset users, we will update the dataset accordingly. The updated version of the dataset will be made available through the dataset release link.

**Will older versions of the dataset continue to be supported/hosted/maintained?**
Yes, with each update, the older versions will remain accessible through their original links.

**If others want to extend/augment/build on/contribute to the dataset, is there a mechanism for them to do so?**
If other researchers or individuals are interested in extending, augmenting, building on, or contributing to the dataset, they should contact us via email, clearly articulating their intentions and requesting our consent prior to any further actions.

## 2 Heuristic algorithm for reading order prediction

The heuristic algorithm employed in Sec. 4.5 of the main paper arranges text lines in a left-to-right, top-to-bottom order, primarily based on the conventional reading order observed in Chinese historical documents. A detailed description of the procedure is provided in Algorithm 1.

---
**Algorithm 1** Sort Bounding Boxes

---
**Input:** $boxes$: a list of bounding boxes to be sorted
**Output:** $sorted\_boxes$: the sorted list of bounding boxes
1: $sorted\_boxes \leftarrow copy(boxes)$
2: $length \leftarrow length(sorted\_boxes)$
3: **for** $i \leftarrow 1$ to $length - 1$ **do**
4:     **for** $j \leftarrow 1$ to $length - i$ **do**
5:         $box1 \leftarrow sorted\_boxes[j]$
6:         $box2 \leftarrow sorted\_boxes[j + 1]$
7:         **if** $box1.center_x < box2.center_x$ **then**
8:             $sorted\_boxes[j] \leftrightarrow sorted\_boxes[j + 1]$
9:         **else if** $box1.center_x = box2.center_x$ **and** $box1.center_y > box2.center_y$ **then**
10:            $sorted\_boxes[j] \leftrightarrow sorted\_boxes[j + 1]$

---

## 3 Additional information of the dataset

### 3.1 Number of text lines/characters

Table 1: Comparison with existing Chinese historical document datasets. * indicates that we only consider the training set due to only the training set of ICDAR 2019 HDRC-CHINESE is available.

| Dataset | Images | Text lines | Characters |
|---|---|---|---|
| MTHv1 [8] | 1,500 | 40,656 | 521,375 |
| MTHv2 [1] | 3,199 | 105,578 | 1,081,663 |
| IC19 HDRC [2]* | 11,715 | 417,489 | 2,482,992 |
| CASIA-AHCDB [3] | - | - | 2,276,740 |
| M$^5$HisDoc (Ours) | 8,000 | 403,824 | 4,367,361 |

As shown in Table 1, $M^5$HisDoc contains 4,367,361 characters, and represents the largest collection in the field of historical document analysis.

## 3.2 Comparison between pre-annotation and manual re-annotation

As mentioned in the Sec. 3.2 in the main paper, we employ models trained on other datasets (MTHv2 [1] and CASIA-AHCDB [3]) for pre-annotation. To enhance the understanding of the extent to which the existing OCR model contributes to the annotation process, we conduct this comparison. That is, i.e., we use manually-labeled labels as GT to evaluate the performance of the models used for pre-annotation. We employ the metrics of character detection and recognition to measure the gap between pre-labeled and manually labeled data. The results are shown in Table 2 and Table 3, respectively. We can see that the models provide about 70% accuracy in bounding box and category labeling of characters.

Table 2: Performance of the character detector in pre-labeling on the manually labeled data.

| IoU thres | Precision↑ | Recall↑ | F1-score↑ |
|-----------|-----------|---------|-----------|
| 0.5 | 96.38 | 85.65 | 90.70 |
| 0.6 | 94.03 | 83.56 | 88.49 |
| 0.7 | 80.76 | 71.77 | 76.00 |

Table 3: Performance of the character recognizer in pre-labeling on the manually labeled data.

| Top-1 acc↑ | Top-5 acc↑ |
|-----------|-----------|
| 76.50 | 84.78 |

# 4 Additional Experiments

## 4.1 Text line/character detection/recognition and reading order prediction

**Setting** The experimental settings in this section align with Sec. 4.1~4.5 of the main paper. We evaluate the performance of different models on the validation set of $M^5$HisDoc.

**Results and analysis** The experimental results are presented in Table 4, 5, 6, 7 and 8, respectively. It can be observed that the experimental findings on the validation set are consistent with those on the test set.

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

## 4.2 Reading order prediction on character-level

**Setting**  We conduct experiments on M$^5$HisDoc for reading order prediction on character-level with both rule-based and learning-based methods. The rule-based method is Augmented XY Cut [31], we reverse the horizontal direction in this algorithm, as Augmented XY Cut is designed for modern documents, which are typically read from left to right. However, the reading order of Chinese historical documents is the opposite. For the learning-based method, we employ LayoutReader [32] for comparison. We utilize Average Relative Distance [32] (ARD) to evaluate the performance.

**Results and analysis**  The experimental results are presented in Table 9. We can observe that the performance of the same method in character-level reading order prediction is significantly inferior compared to the text lines-level discussed in Sec. 4.5 of the main paper. This indicates that predicting the reading order at character-level presents a significant challenge, thus highlighting it as a promising research direction for future investigations.

Table 9: Results of reading order prediction on character-level, in the format of M$^5$HisDoc-H/M$^5$HisDoc-R test set. ARD: Average Relative Distance.

| Type | Method | Venue | ARD↓ |
|---|---|---|---|
| Rule-based | Augumented XY Cut [31] | CVPR'22 | 193.16/160.97 |
| Learning-based | LayoutReader [32](layout only) | EMNLP'21 | 51.72/37.97 |

## 4.3 Cross-validation between M$^5$HisDoc-R and M$^5$HisDoc-H

**Setting** We perform cross-validation between M$^5$HisDoc-R and M$^5$HisDoc-H using the Cascade R-CNN [10], CRNN [17], YOLOX [25], and ConvNeXt [28] models mentioned in Sec. 4.1~4.4 of the main paper.

Table 10: Cross-validation between the models trained on M$^5$HisDoc-R and M$^5$HisDoc-H. The evaluation metric for text line/character detection is F-score at 0.5 IoU, the metrics for text line/character recognition are AR and top-1 accuracy, respectively.

| Tasks | Text line det (F1-score) | Text line reg (AR) | Char det (F1-score) | Char reg (top-1 acc) |
|---|---|---|---|---|
| M$^5$HisDoc-R → M$^5$HisDoc-H | 81.89 | 87.48 | 79.24 | 93.39 |
| M$^5$HisDoc-H → M$^5$HisDoc-R | 95.39 | 90.78 | 98.35 | 94.94 |

**Results and analysis** The results are presented in Table 10, it can be observed that the models trained on M$^5$HisDoc-H outperforms the one trained on M$^5$HisDoc-R in cross-validation. These results highlight the significance of incorporating datasets such as M$^5$HisDoc-H in addressing the intricate challenges associated with historical document analysis, particularly in complex scenarios.

## 5 Computational Resources

All experiments in this study are conducted on four RTX 3090 GPUs.

## 6 Illustration of model output

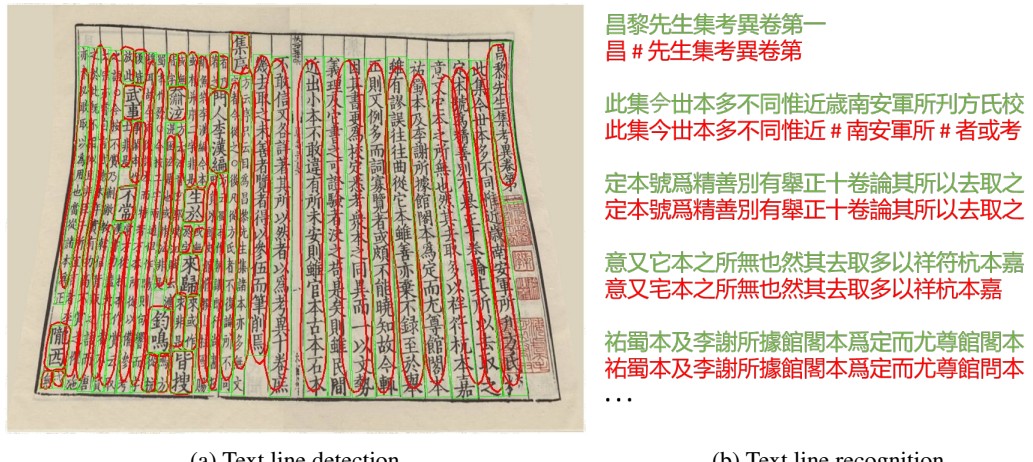

(a) Text line detection          (b) Text line recognition

Figure 2: Illustration of model output. The red polygons and text represent the output of the model and the green ones represent GT.

# 7 Additional visual illustrations of M⁵HisDoc

## 7.1 Illustration of the data source

As mentioned in Sec. 3.1 in the main paper, our data source from MTHv2 (Fig. 3a), SCUT-CAB (Fig. 3b), pictures we took (Fig. 3c), and the Internet (Fig. 3d).

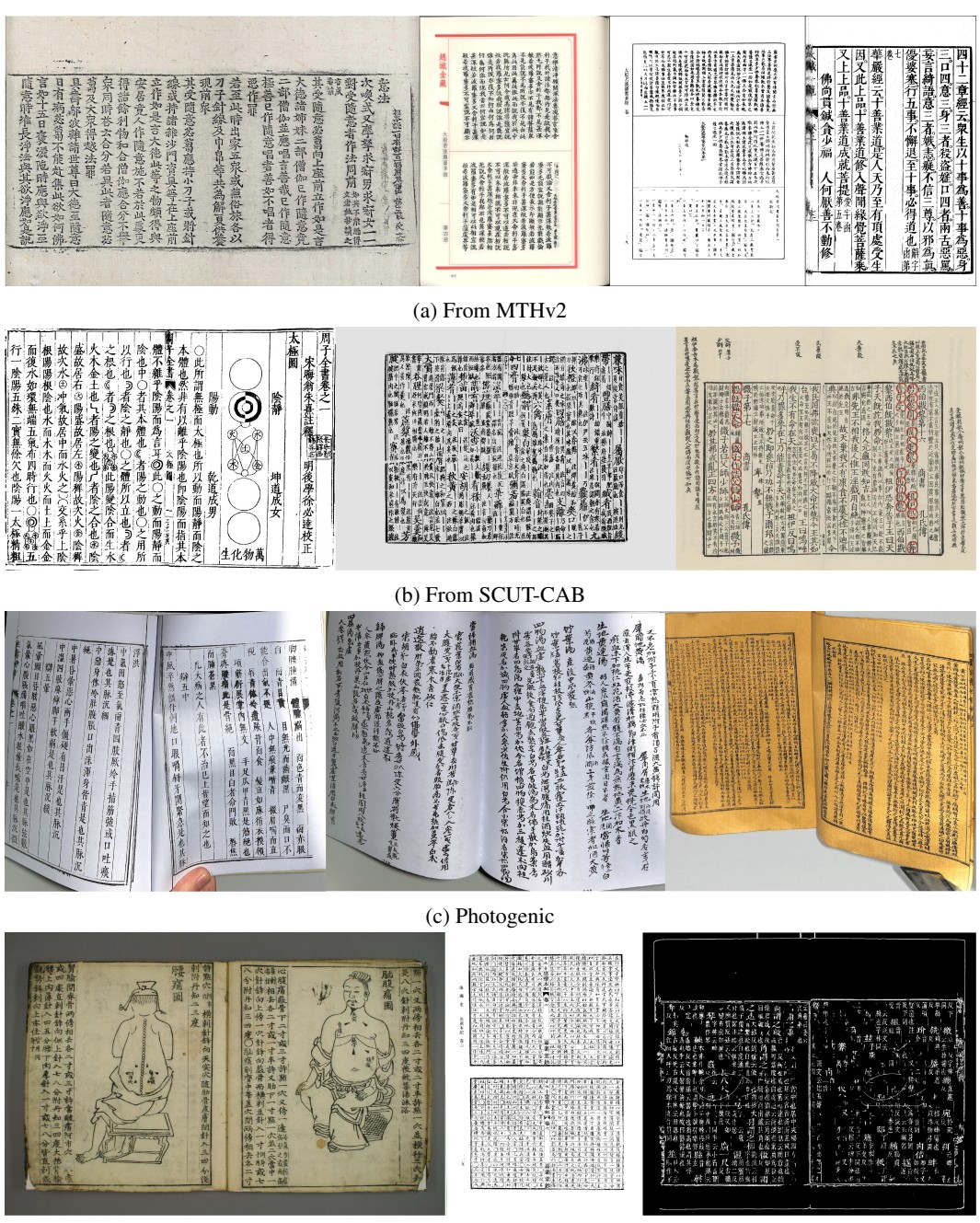

(a) From MTHv2

(b) From SCUT-CAB

(c) Photogenic

(d) From Internet

Figure 3: Illustration of the data source.

## 7.2 Illustration of various calligraphic styles

As described in Sec. 3.3 of the main paper, the M$^5$HisDoc contains various calligraphic styles, including regular, clerical, running, and cursive scripts (As shown in Fig. 4, respectively). These categories differ in Chinese calligraphy, each characterized by its unique writing style. For example, cursive script presents a more casual and flowing writing style, usually resulting in less recognizable characters, while regular script exhibits a more standardized and formal typeface.

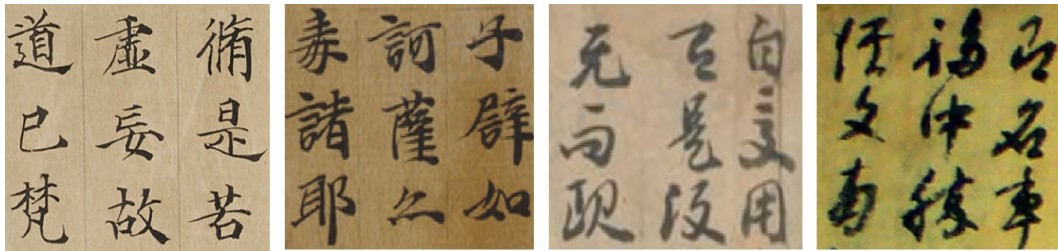

Figure 4: Illustration of various calligraphic styles.

## 7.3 Illustration of various layout types

"Layout type" refers to the various types of text arrangements and sizes within a document. This includes variations such as text divided into double blocks (left, right), text segregated into three parts (top, middle, and bottom), etc. Diverse layouts significantly influence text detection and reading order. As shown in Fig. 5, there are three different types of layout.

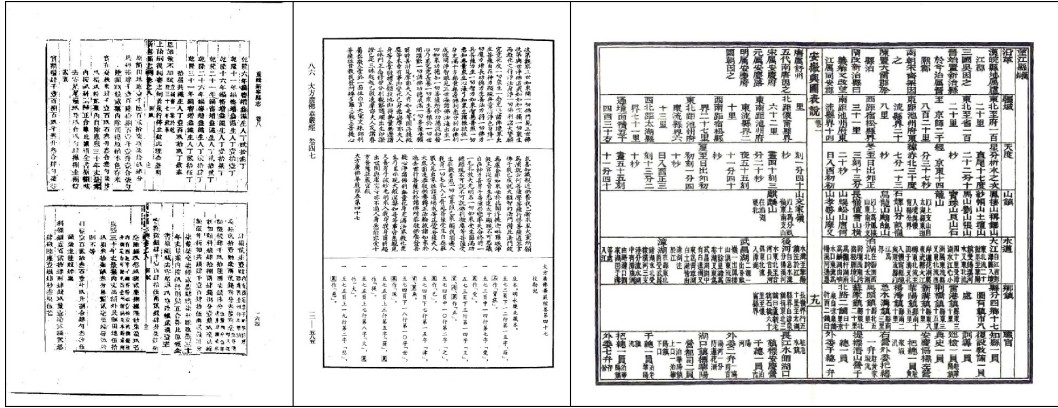

Figure 5: Illustration of various layout types.

## 7.4 Various features of M$^5$HisDoc

Fig. 6, 7, 8, 9, 10, and 11 demonstrate the features blurry texts, complex handwritten texts, variations in font sizes, complex arrangement of text content, dense texts, and distortion of M$^5$HisDoc, respectively.

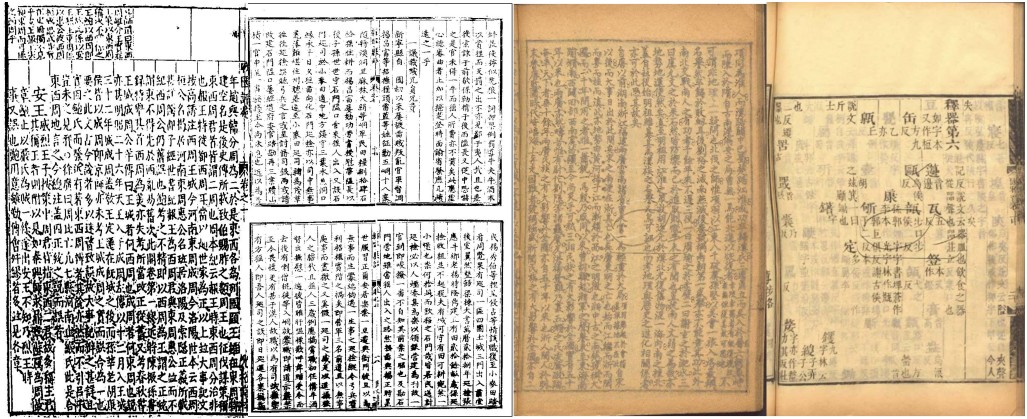

Figure 6: Blurry texts.

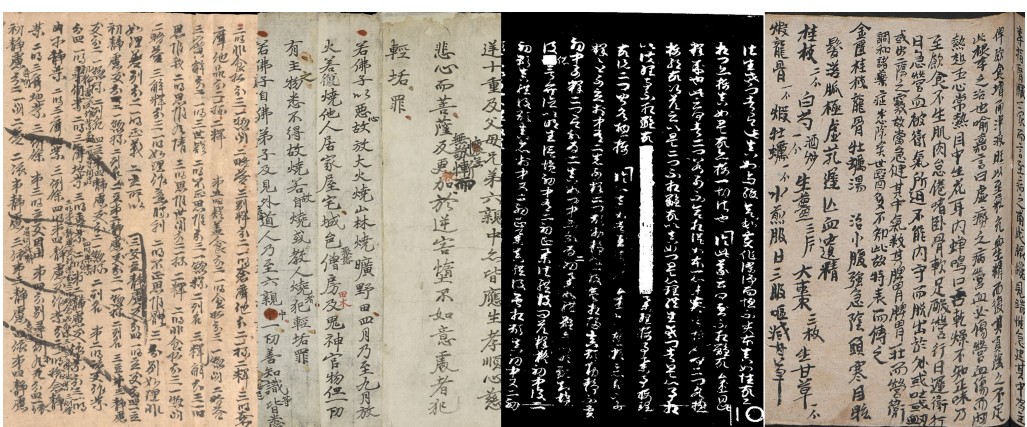

Figure 7: Complex handwritten texts.

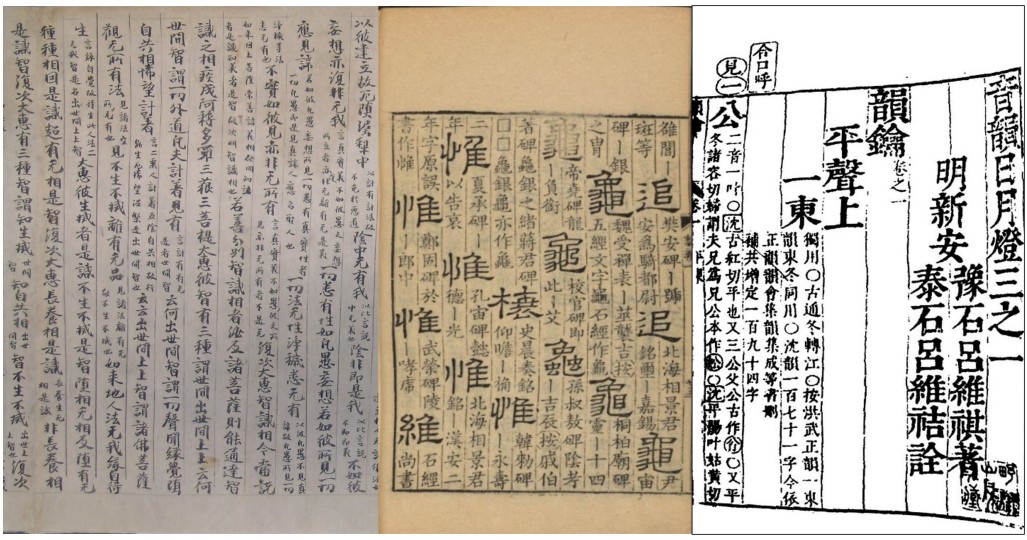

Figure 8: Variations in font sizes.

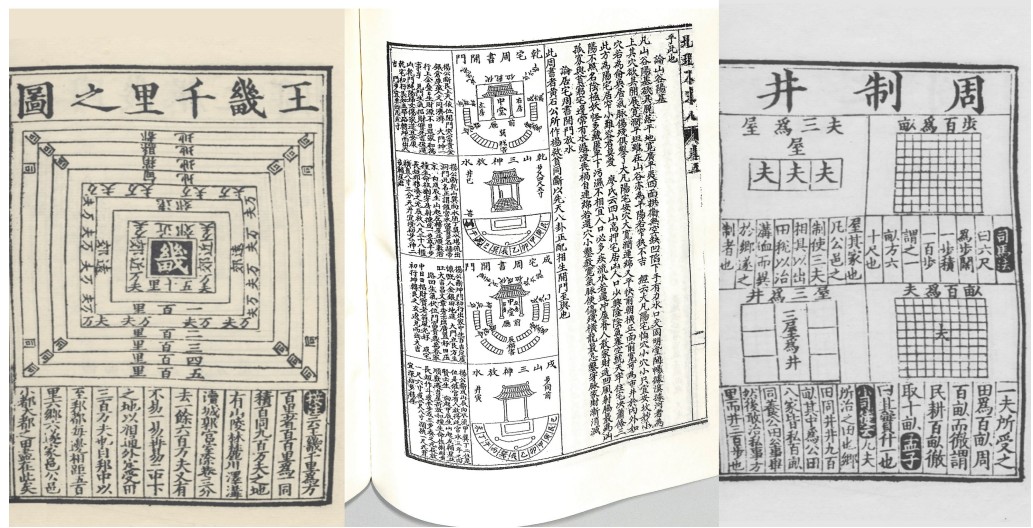

Figure 9: Complex arrangement of text content.

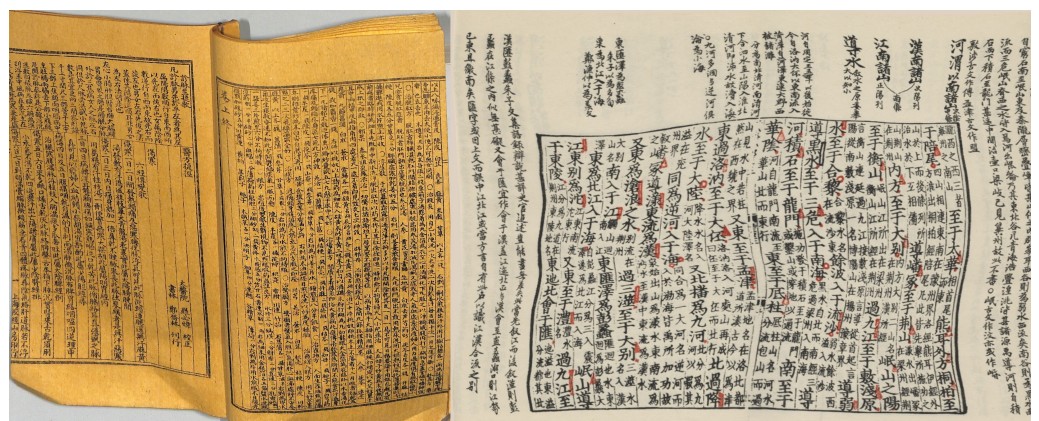

Figure 10: Dense texts.

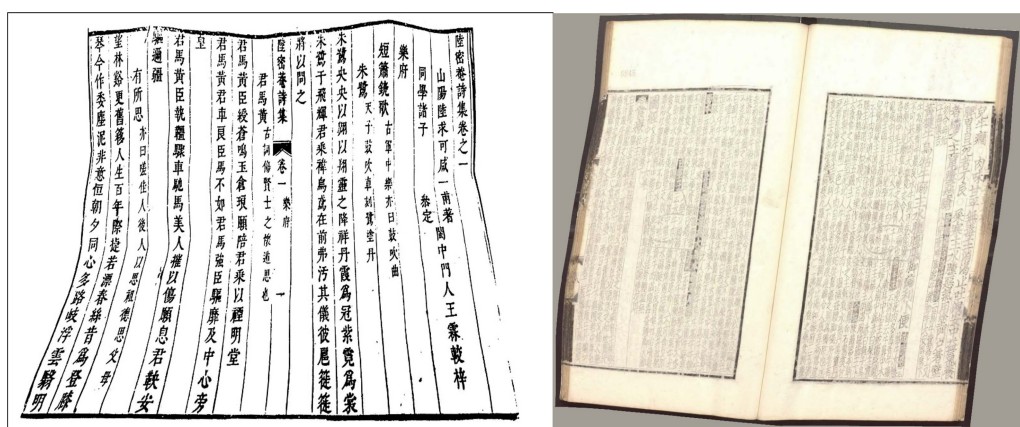

Figure 11: Distortion.