# OpenReview forum: "M5HisDoc: A Large-scale Multi-style Chinese Historical Document Analysis Benchmark"
_NeurIPS.cc/2023/Track/Datasets_and_Benchmarks — NeurIPS 2023 Datasets and Benchmarks Poster_

### Official Review · Reviewer_mes1 · 2023-06-28
**M5HisDoc: A Large-scale Multi-style Chinese Historical Document Analysis Benchmark**

**Rating:** 7
**Confidence:** 4
**Clarity:** The paper is well written.

**Strengths:**

- The paper proposed a new Chinese historical document dataset for various historical document image analysis task.
- Thorough and extensive experiential benchmark to provide baseline results using $M^5$HisDoc dataset.

**Additional Feedback:**

-

**Correctness:**

The data collection, preparation process, and evaluation methods are well described. However, it would be beneficial if the authors could review the Chinese historical dataset that was previously created, e.g https://arxiv.org/pdf/1903.03341.pdf, and provide a clear comparison between their dataset and the previous one.  A detailed and objective comparison would enhance the credibility of their findings.

**Documentation:**

For the dataset collection and organization, it is clear; and also provided detail description about the data access and usage in the supplementary material. But very limited description is given from their Github repository.

**Ethics:**

There is no any notable ethical concern with this work.

**Limitations:**

The authors state the limitation of their work in the supplementary material, and this research work does not exhibit any notable negative societal impact.

**Opportunities For Improvement:**

- The paper does not clearly state the dates of historical document production that were considered during the data collection process.
- It would be beneficial if the author provided a comparison between the predictions of the OCR model (trained on older datasets i.e  MTHv2 [5] and CASIA-AHCDB [7]) and the annotations made by domain expert annotators. This would enhance our understanding of the extent to which the existing OCR model contributes to the annotation process.
- The authors attempted to justify the quantitative differences between their new dataset and the existing dataset, as explained in their experimental section. However, I am curious to understand how such a significant performance gap was observed. Additionally, I am uncertain why the authors considered the CASIA-AHCDB [7] dataset in the previous work as lacking diversity. This dataset comprises images from more than 37 books for the training set and over 12 books for the test set  (see section A of (refer to https://arxiv.org/pdf/1903.03341.pdf or  Table 1 of https://sais2019.cs.umu.se/wp-content/uploads/2019/06/SAIS_2019_paper_18.pdf). Moreover, this dataset contains other feature and over 8,000 unique characters.
- Better to provide how the 300 images from MTHv2 [5] and 700 images from SCUT-CAB [11] are selected i.e what kind of document images are selected?
- The rationale behind the author's decision to focus on character-level annotation/recognition, despite line-level or above being sufficient due to the state-of-the-art segmentation-free OCR methods, is unclear.

**Relation To Prior Work:**

The author discusses the differences between this dataset and the related dataset introduced in previous works but overlooks some important features that were addressed in the previous dataset e.g diversity of document images, number of unique character, etc.

**Summary And Contributions:**

This authors propose a new chinese histotical document benchmarks named $M^5$HisDoc for historical document analysis task including textline/character detection and recogntion and reading order. that comprises aboutr 8,000 multi-style images. The dataset consists of two subsets, $M^5$HisDoc-R (Regular) and $M^5$HisDoc-H (Hard).

---

> ### Author Response · Authors · 2023-08-12
> **Response to reviewer mes1 (part 1)**
>
> Thank you for your constructive comments and suggestions. In the following, your comments are first stated and then followed by our point-by-point responses. Due to the character limit, the response is divided into two parts.
>
> ---
>
> > **Q1**:  The paper does not clearly state the dates of historical document production that were considered during the data collection process.
>
>
> **A1**:  During the data collection process, we made an effort to gather historical documents from various periods. The final dataset includes historical documents spanning different periods, such as the Han (206 BCE – 220 CE), Tang (618 – 907 CE), Song (960 – 1279 CE), Ming (1368 – 1644 CE), and Qing (1644 – 1912 CE) dynasties. We will add this information to the main text.
>
>
> > **Q2**:  It would be beneficial if the author provided a comparison between the predictions of the OCR model (trained on older datasets i.e MTHv2 [5] and CASIA-AHCDB [7]) and the annotations made by domain expert annotators. This would enhance our understanding of the extent to which the existing OCR model contributes to the annotation process.
>
>
> **A2**:  Thanks for your suggestion. We greatly acknowledge the importance of this comparison, and we intend to include it in the final version.
>
>
> > **Q3**:  The authors attempted to justify the quantitative differences between their new dataset and the existing dataset, as explained in their experimental section. However, I am curious to understand how such a significant performance gap was observed. Additionally, I am uncertain why the authors considered the CASIA-AHCDB [7] dataset in the previous work as lacking diversity. This dataset comprises images from more than 37 books for the training set and over 12 books for the test set (see section A of (refer to https://arxiv.org/pdf/1903.03341.pdf or Table 1 of https://sais2019.cs.umu.se/wp-content/uploads/2019/06/SAIS_2019_paper_18.pdf). Moreover, this dataset contains other feature and over 8,000 unique characters.
>
>
> **A3**:
> 1. The reason for the significant disparity is that previous datasets largely focused on specific document types, such as MTHv2 containing only Buddhist scriptures and IC19 HDRC containing only genealogical records. In contrast, our M^5HisDoc tried to collect a diverse range of document types as extensively as possible during the data collection.
>
> 2. The diversity discussed in the paper largely refers to the page-level. For CASIA-AHCDB, it is a dataset at the character-level, which is not suitable for text line and page level tasks such as text line detection and recognition. And we do not consider that it lacks diversity at the character-level. On the contrary, we view this dataset as exhibiting diversity at the character-level (contains various styles and a wide range of character categories). This is why we utilized it to train a recognition model for preliminary annotation, which is crucial for reducing the workload of annotators.
>
>
> > **Q4**:  Better to provide how the 300 images from MTHv2 [5] and 700 images from SCUT-CAB [11] are selected i.e what kind of document images are selected?
>
>
> **A4**:  The images from MTHv2 and SCUT-CAB were manually selected under the criteria that the selected images possess certain characteristics (such as dense text or complex layout). We will include this information in the final version.
>
>
> > **Q5**:  The rationale behind the author's decision to focus on character-level annotation/recognition, despite line-level or above being sufficient due to the state-of-the-art segmentation-free OCR methods, is unclear.
>
>
> **A5**:  The reasons for our focus on character-level annotation are primarily twofold.
>
> 1. Our objective is to offer annotation data across multiple levels, enabling researchers in the field of historical document analysis to explore various approaches.
>
> 2. As shown in Tables 2 and 4, for some complex scenarios, the character-level approaches exhibit superior robustness and accuracy.

---

> > ### Author Response · Authors · 2023-08-12
> > **Response to reviewer mes1 (part 2)**
> >
> > > **Q6**:  The data collection, preparation process, and evaluation methods are well described. However, it would be beneficial if the authors could review the Chinese historical dataset that was previously created, e.g https://arxiv.org/pdf/1903.03341.pdf, and provide a clear comparison between their dataset and the previous one. A detailed and objective comparison would enhance the credibility of their findings.
> >
> >
> > **A6**:  Thanks for your suggestion, we will add this information in the final version.
> >
> >
> > > **Q7**:  The author discusses the differences between this dataset and the related dataset introduced in previous works but overlooks some important features that were addressed in the previous dataset e.g diversity of document images, number of unique character, etc.
> >
> >
> > **A7**:  We will revise the relevant descriptions to ensure that we do not overlook any important features of previous datasets. However, it is safe to say that our dataset presents even more significant challenges. For instance, M^5HisDoc includes over 20 types of historical documents and an astonishing 16,151 character categories, which is considerably larger than those found in existing datasets.
> >
> > > **Q8**:  For the dataset collection and organization, it is clear; and also provided detail description about the data access and usage in the supplementary material. But very limited description is given from their Github repository.
> >
> >
> > **A8**:  We have added more descriptions to the GitHub repository as per your suggestion.

---

> > > ### Comment · Reviewer_mes1 · 2023-08-22
> > >
> > > The author has addressed my comment partially, and given the paper's current state, I will retain my original score of 7.

---

### Official Review · Reviewer_5Ty8 · 2023-07-18
**M5HisDoc: A Large-scale Multi-style Chinese Historical Document Analysis Benchmark**

**Rating:** 8
**Confidence:** 5
**Correctness:** The claims made in the paper are corr…
**Clarity:** Well written

**Strengths:**

(1) The paper is well written and the motivation is made clear.

(2) The workload is significant. The entire data collection, annotation, and check process takes approximately 4,000 person-hours.

(3) The M5 indicates five properties of style, ie., Multiple layouts, Multiple document types, Multiple calligraphy styles, Multiple backgrounds and Multiple challenges.

(4) Thorough baseline experiments are conducted.


**Additional Feedback:**

See Limitations

**Documentation:**

The download link is invalid.

**Ethics:**

No ethical concerns

**Limitations:**

(1) From the experimental results, all tasks achieved good performance on the proposed M5HisDoc. I suspect the reason why the experimental results are so good is because some of the images in the training and test sets are from the same ancient book. Therefore, authors are encouraged to conduct experiments to demonstrate the universality of the model trained using this dataset, that is, apply the model to the unseen ancient books, or divide the training and test sets by book instead of image.

(2) The download link provided from dataset Url ‘https://github.com/HCIILAB/M5HisDoc’ is invalid.


**Opportunities For Improvement:**

See Limitations

**Relation To Prior Work:**

Sufficient

**Summary And Contributions:**

This paper proposes a complex multi-style 57 Chinese historical document analysis benchmark, named M5HisDoc, which contains 8,000 images in 58 multiple styles. The benchmark exhibits feature of large scale, high text density, extensive character 59 set, and diverse styles. Authors also conduct thorough baseline experiments using M5HisDoc, and demonstrate M5HisDoc presents novel challenges for historical document analysis.

---

> ### Author Response · Authors · 2023-08-12
> **Response to reviewer 5Ty8**
>
> Thank you for your positive feedback and insightful review. Here are the responses to your comments.
>
> ---
>
> > **Q1**: From the experimental results, all tasks achieved good performance on the proposed M5HisDoc. I suspect the reason why the experimental results are so good is because some of the images in the training and test sets are from the same ancient book. Therefore, authors are encouraged to conduct experiments to demonstrate the universality of the model trained using this dataset, that is, apply the model to the unseen ancient books, or divide the training and test sets by book instead of image.
>
>
> **A1**:  As you might guess, some of the images in the training and test sets are from the same ancient book. To demonstrate the generalization of the model trained using this dataset, we have included cross-validation experiments with IC19 HDRC and CASIA-AHCDB in the supplementary material (Section 5.3). The experimental results indicate that models trained on M^5HisDoc exhibit better generalization performance compared with other datasets.
>
>
> > **Q2**: The download link provided from dataset Url ‘https://github.com/HCIILAB/M5HisDoc’ is invalid.
>
>
> **A2**:  Thank you for pointing out this issue. The original download link on Baidu Drive was mistakenly deleted due to a system error. We have updated the download link on Baidu Drive (https://pan.baidu.com/s/1PSyADEe4cIa0zxS2erQCfA?pwd=c2y1). Additionally, to avoid the reoccurrence of similar issues, we will conduct regular checks to ensure the validity of the links.

---

### Official Review · Reviewer_eX73 · 2023-07-18
**M5HisDoc benchmark dataset**

**Rating:** 7
**Confidence:** 3

**Strengths:**

- Meticulous annotation procedure (Section 3.2 first paragraph). Some readers/reviewers might want to see statistics on how many errors were fixed by this process, inter-rater agreement scores, etc.
- The dataset fills a gap in this research space.
- The dataset is more diverse (by several meanings of the word) than alternative resources.

**Additional Feedback:**

see questions above

**Clarity:**

-  Some minor grammar issues. For example line 72 should be "Chinese historical documents" (plural) rather than "Chinese historical document" (singular). Section 2.1 header should be "Historical document benchmarks" and not "Historical document benchmark".
- Q1) It is unclear to me what "Cls" is in Table 1.
- Q2) It is unclear to me what the precise definition of "layout type" is in the context of the discussion in Section 3.3.
- Q3) Similarly, how were calligraphic styles determined in Section 3.3?
- Q4) Why was MTHv2 the only dataset used in Section 4.6?

**Correctness:**

- See Q1 - Q6 below.

**Documentation:**

- Q5) Are the sources of data (e.g., "we gather tens of thousands of scanned images 130 from electronic ancient books available on the Internet" in Section 3.1) listed anywhere?
- Q6) Similarly, it is good practice to mention if there are any licenses associated with the data sources. Since the documents are historical, though, it is unlikely that this is a problem (e.g., with copyright law).

**Limitations:**

- The idea of including the harder -R challenge dataset is commendable, but the distortions are somewhat limited and are not given much motivation. I wonder if a tailored tool like Augraphy (https://github.com/sparkfish/augraphy) could be used to produce more distortions.
- The dataset is limited to Chinese. As a non-expert, I wonder if there is a similar lack of historical datasets for other languages as well.

**Opportunities For Improvement:**

- More clarity around the different tasks would help readers. The tasks discussed in the paper are text line detection, text line recognition, character detection, character recognition, and reading order prediction. I had trouble understanding the difference between text line detection and text line recognition, and the paper assumes the reader is already familiar with these two concepts.
- Similarly, more discussion on the evaluation metrics would be helpful to non-experts. What is the intuition behind 1-NED, CR, AR, ARD, etc.?
- Examples of model output (especially showing errors alongside ground-truth annotations) for the various tasks would be helpful.
- Some clarity around the historical-ness of the data would be helpful. I find it a little surprising that a paper on historical documents wouldn't mention what years or historical time periods these documents are from.

**Relation To Prior Work:**

- Prior work on reading order prediction is not included in the Related Work section, but prior work on other tasks is included.

**Summary And Contributions:**

This paper introduces a new dataset for historical document processing dataset called M5HisDoc. The dataset is meticulously annotated for several tasks: text line detection & recognition, character detection & recognition, and reading order prediction. The dataset fills a gap in the research space by providing a diverse, large dataset for these tasks. Many (I assume) contemporary models are evaluated and benchmarked on M5HisDoc. Model performance scores on the new M5HisDoc are often low, indicating that the new dataset will help serve as a valuable benchmark going forward.

---

> ### Author Response · Authors · 2023-08-12
> **Response to reviewer eX73 (part 1)**
>
> Thank you for your detailed review and constructive suggestions. Point-by-point responses to your comments are as follows. Due to the character limit, the response is divided into three parts.
>
> ---
>
> > **Q1**:  It is unclear to me what "Cls" is in Table 1.
>
>
> **A1**:  "Cls" in Table 1 is an abbreviation for "Class", representing "The number of character categories". Due to the extensive content of the table, we have employed its abbreviation in the field.
>
> In the final version, we will provide an explanation for this abbreviation.
>
>
> > **Q2**:  It is unclear to me what the precise definition of "layout type" is in the context of the discussion in Section 3.3.
>
>
> **A2**:  "Layout type" refers to the various types of text arrangements and sizes within a document.
> This includes variations such as text divided into double blocks (left, right), text segregated into three parts (top, middle, and bottom), etc.
> Diverse layouts significantly influence the text detection and reading order.
>
> In the final version, we will provide some illustrations to facilitate the comprehension of the various layout types.
>
>
> > **Q3**:  Similarly, how were calligraphic styles determined in Section 3.3?
>
>
> **A3**:  As described in line 170 of the main text, the M^5HisDoc contains various calligraphic styles, including cursive, clerical, regular, and running scripts. These categories differ in Chinese calligraphy, each characterized by its unique writing style. For example, cursive script presents a more casual and flowing writing style, usually resulting in less recognizable characters, while regular script exhibits a more standardized and formal typeface.
>
> In the final version, we will provide some illustrations to enhance the comprehension of the diverse calligraphic styles.
>
>
> > **Q4**:  Why was MTHv2 the only dataset used in Section 4.6?
>
>
> **A4**:  MTHv2 is the only available dataset with both character-level and text-line-level annotations, which enables us to conduct cross-validation experiments at both levels of tasks. Additionally, we also conducted cross-validation experiments on other datasets (IC19 HDRC and CASIA-AHCDB), and the results were included in the supplementary material (Section 5.3).
>
>
> > **Q5**:  Are the sources of data (e.g., "we gather tens of thousands of scanned images 130 from electronic ancient books available on the Internet" in Section 3.1) listed anywhere?
>
>
> **A5**:  Thanks for your suggestion. We will provide this information.
>
>
> > **Q6**:  Similarly, it is good practice to mention if there are any licenses associated with the data sources. Since the documents are historical, though, it is unlikely that this is a problem (e.g., with copyright law).
>
>
> **A6**:  The historical documents within our dataset are in the public domain, as their authors have been deceased for over 50 years. Additionally, the images we obtained from the Internet are sourced from some websites that are under open-copyright licenses for non-commercial purposes, thereby avoiding the copyright issues.

---

> > ### Author Response · Authors · 2023-08-12
> > **Response to reviewer eX73 (part 2)**
> >
> > > **Q7**:  More clarity around the different tasks would help readers. The tasks discussed in the paper are text line detection, text line recognition, character detection, character recognition, and reading order prediction. I had trouble understanding the difference between text line detection and text line recognition, and the paper assumes the reader is already familiar with these two concepts.
> >
> >
> > **A7**:  Text line detection and text line recognition are two popular research topics in the field of optical character recognition (OCR) and computer vision in recent years. A brief review of some typical methods for text line detection and recognition was given in Sections 2.2 and 2.3 of the main text. We also give a brief introduction to these two terms below.
> >
> > 1. Text Line Detection: The process of locating the text regions of document images. The output of this task is typically a set of bounding boxes or polygons surrounding each detected text line.
> >
> > 2. Text Line Recognition: The process of recognizing the text content including characters, numbers, and symbols within detected text line regions. The output of this task is typically a machine-readable text string for each detected text line.
> >
> >
> > > **Q8**:  Similarly, more discussion on the evaluation metrics would be helpful to non-experts. What is the intuition behind 1-NED, CR, AR, ARD, etc.?
> >
> >
> > **A8**: Below is the introduction to 1-NED, CR, AR, and ARD.
> > 1. 1-NED:
> >     This metric evaluates the end-to-end performance of text line detection and recognition. It first matches the text lines predicted by the detection model with the ground truth (GT), penalizes unmatched text lines, and then calculates the textual similarity between the content predicted by the recognition model and GT within each matched pair.
> >
> >     The formula is:
> >
> >     $NED\left(s_1,s_2\right)=edit\_dist\left(s_1,s_2\right)/max\left(l_1,l_2\right)$
> >     $1-NED={\textstyle\sum_{i=1}^n}NED\left(s_{i1},s_{i2}\right)/n$
> >
> >     where $edit\_dist()$ refers to the minimum number of editing operations required to transform one string into another one between two strings, $s_1$ and $s_2$ are the text strings of a matching pair, $l_1$, $l_2$ are their text lengths, and $n$ is the total number of text lines.
> >
> >
> > 2.  CR、AR:
> >     These two metrics are used to measure the text line recognizer's performance by comparing the similarity between the text recognized by the model and the GT.
> >
> >     The formula is:
> >
> >     $AR = \left(N_t-D_e-S_e-I_e\right)/N_t$
> >     $CR = \left(N_t-D_e-S_e\right)/N_t$
> >
> >     where $D_e$, $S_e$, and $I_e$ represent the total number of deletion, substitution, and insertion errors, respectively, the errors are calculated between the recognition result and GT, $N_t$ is the total number of characters in the annotations.
> >
> >
> > 3. ARD:
> >     This metric measures the relative distance between common elements in different sequences. It is used to compare the predicted reading order sequence with the GT sequence.
> >
> >     The formula is:
> >
> >     $s(e_k, B) =
> >     \begin{cases}
> >     |k - I(e_k, B)|, & \text{if } e_k \in B \\
> >     n, & \text{otherwise}
> >     \end{cases}$
> >
> >     $ARD(A, B) = \frac{1}{n} \sum_{e_k \in A} s(e_k, B)$
> >
> >     where $e_k$ is the k-th element in sequence $A$, $I\left(e_k,B\right)$ is the index of $e_k$ in sequence $B$, and $n$ is the length of sequence $A$.
> >
> >
> > > **Q9**:  Examples of model output (especially showing errors alongside ground-truth annotations) for the various tasks would be helpful.
> >
> >
> > **A9**:  Thanks for your suggestion. We will include additional examples in the final version.
> >
> >
> > > **Q10**:  Some clarity around the historical-ness of the data would be helpful. I find it a little surprising that a paper on historical documents wouldn't mention what years or historical time periods these documents are from.
> >
> >
> > **A10**:  During the data collection process, we made an effort to gather historical documents from various periods. The final dataset includes historical documents spanning different periods, such as the Han (206 BCE – 220 CE), Tang (618 – 907 CE), Song (960 – 1279 CE), Ming (1368 – 1644 CE), and Qing (1644 – 1912 CE) dynasties.

---

> > > ### Author Response · Authors · 2023-08-12
> > > **Response to reviewer eX73 (part 3)**
> > >
> > > > **Q11**:  The idea of including the harder -R challenge dataset is commendable, but the distortions are somewhat limited and are not given much motivation. I wonder if a tailored tool like Augraphy (https://github.com/sparkfish/augraphy) could be used to produce more distortions.
> > >
> > >
> > > **A11**:  Thank you for sharing this tool. It does indeed generate more distortions. However, we encountered a challenge when applying this tool to image deformations. Specifically, we are unable to obtain text position annotations (bounding boxes of text lines and characters) for the deformed images. In our current approach, we are able to derive the transformed text positions through coordinate mapping. Furthermore, the distortions we employed are based on deformation fields extracted from actual photographed images, enabling us to simulate distortions in real-world scenarios more accurately.
> > >
> > >
> > > > **Q12**:  The dataset is limited to Chinese. As a non-expert, I wonder if there is a similar lack of historical datasets for other languages as well.
> > >
> > >
> > > **A12**:  As mentioned in Section 2.1 of the main text, there are some Historical Document benchmarks available in other languages. However, to the best of our knowledge, there is a similar lack of historical datasets for these languages.
> > >
> > >
> > > > **Q13**:  Some minor grammar issues. For example line 72 should be "Chinese historical documents" (plural) rather than "Chinese historical document" (singular). Section 2.1 header should be "Historical document benchmarks" and not "Historical document benchmark".
> > >
> > >
> > > **A13**:  Thank you for pointing out some grammar issues for us. We will carefully review the paper and make the necessary corrections to address these problems.
> > >
> > >
> > > > **Q14**:  Prior work on reading order prediction is not included in the Related Work section, but prior work on other tasks is included.
> > >
> > >
> > > **A14**:  The reading order prediction aims to arrange the text within a document in the correct order for reading. LayoutReader [54] is a sequence-to-sequence model that captures both text and layout information to predict the reading order. Gu et al. proposed a heuristic algorithm, Augmented XY Cut [55], to sort the input tokens.
> > >
> > > We will include the above introduction in the "Related Works" section of the main text.

---

### Official Review · Reviewer_oBLA · 2023-07-22
**A bottomline case**

**Rating:** 6
**Confidence:** 3
**Clarity:** The presentation can be more informat…

**Strengths:**

1) The proposed data set seems to be more diverse in multiple aspects compared to existing data sets.

2) Multiple modeling tasks are evaluated on the data set.

**Additional Feedback:**

See "Opportunities For Improvement" section.

**Correctness:**

I am not sure whether the proposed data set can be a good evaluation set before showing its results correlate well with data sets from physical books.

**Documentation:**

Some details are missing, especially the details of the proposed data set.

**Ethics:**

No major ethics issue.

**Limitations:**

No limitations are discussed in this paper, though some limitations should be, like the one in the above section.

**Opportunities For Improvement:**

1) The presentation can be more informative. For example, the data set consist of three categories: existing data sets like MTHv2, scanned images from electronic books, phot shoots of physical books. Can the authors provide examples for each category. 2) For the post processing operations, can the authors provide some examples for before/after the operators so that it will be intuitive to understand how the post-processing looks like.

2) The real ground truth for historical document analysis will be photo copies from the physical books. The proposed data set are more like an augmented training set instead of a evaluation set. To convince the proposed data set is useful for evaluation, the authors should prove the evaluation results on it can well represent the results on other evaluation sets from physical books.

3) The cross-validation experiments are interesting, however, since M^5HisDoc-H contains some MTHv2 data, it might not be surprising that the models trained on  M^5HisDoc-H works better on MTHv2 than the model trained on MTHv2 and tested on M^5HisDoc-H. The experiments are more interesting if MTHv2 is replaced by another data sets which is not included in M^5HisDoc-H at all.

**Relation To Prior Work:**

Some comparison to existing historical document data sets are provided in Table 1.

**Summary And Contributions:**

This paper proposed a benchmark for Chinese historical document analysis. The data collection consists of three parts: 1) images from MTHv2 and SCUT-CAB 2) scanned images from 131 electronic ancient books 3) photo shoots of 4 physical Chinese ancient books. Multiple annotations are conducted including adjust inaccurate character bounding boxes, proof character recognition results, determine which characters belong to the same line, and sort the text lines in correct reading order. Then some post-processing and augmentation are applied including rotation, distortion by DewarpNet, etc. The authors claim the proposed data sets M^5: multiple layouts, document types, calligraphy styles, backgrounds, and multiple challenges.

Several modeling tasks including text line detection, text line recognition, character detection and recognition, reading order prediction are evaluated with multiple models for each task. The authors provide a cross-validation experiment to train models on M5HisDoc-H and test on MTHv2, and vice versa.

---

> ### Author Response · Authors · 2023-08-12
> **Response to reviewer oBLA**
>
> Thank you for your detailed and thoughtful review. Here are our responses to your concerns.
>
> ---
>
> > **Q1**: The presentation can be more informative. For example, the data set consist of three categories: existing data sets like MTHv2, scanned images from electronic books, phot shoots of physical books. Can the authors provide examples for each category. 2) For the post processing operations, can the authors provide some examples for before/after the operators so that it will be intuitive to understand how the post-processing looks like.
>
>
> **A1**:  We have provided some examples (Figure 2) as well as some representative images (Figure 3 - Figure 8) in the supplementary material. Based on your suggestion, we will incorporate some of the examples into the main text (Fortunately, the camera-ready version can accommodate an additional page of content).
>
>
> > **Q2**: The real ground truth for historical document analysis will be photo copies from the physical books. The proposed data set are more like an augmented training set instead of a evaluation set. To convince the proposed data set is useful for evaluation, the authors should prove the evaluation results on it can well represent the results on other evaluation sets from physical books.
>
>
> **A2**:  As presented in Section 3 of the main text, M^5HisDoc consists of two subsets: M^5HisDoc-R and M^5HisDoc-H.
>
> 1. The images contained within M^5HisDoc-R are sourced from physical books, the evaluation results on it can well represent the results on other evaluation sets from physical books.
>
> 2. M^5HisDoc-H is created based on M^5HisDoc-R. We applied corresponding post-processing on images in M^5HisDoc-R to better simulate the challenges introduced during photography, including rotation, distortion, and low resolution.
>
> 3. To demonstrate the significance of M^5HisDoc-H, we conduct the subsequent experiment:
> Firstly, we create a new test set, which consists of 200 camera-captured images of physical books.
> Subsequently, we evaluate the Cascade R-CNN, CRNN, YOLOX, and ConvNeXt models which were trained on M^5HisDoc-R/H and mentioned in Section 4.1 - Section 4.4 of the main text.
> The evaluation metric for text line/character detection is F1-score with an IoU threshold of 0.7, and the metrics for text line and character recognition are AR and top-1 accuracy. The results of the experiment are listed in the below table.
>
> | Traing set | Text line det (F1-score) | Text line recog (AR) | Char det (F1-score) | Char recog (Top-1 acc)|
> |:-------:|:-------:|:-------:|:-------:|:-------:|
> | M^5HisDoc-R | 97.12 | 90.43 | 91.31 | 94.25 |
> | M^5HisDoc-H | **97.25** | **91.63** | **95.19** | **94.42** |
>
> As shown in the table above, the results show that the models trained on M^5HisDoc-H exhibit better performance. This substantiates the capability of M^5HisDoc-H to accurately simulate the challenges introduced during photography.
>
>
> > **Q3**: The cross-validation experiments are interesting, however, since M^5HisDoc-H contains some MTHv2 data, it might not be surprising that the models trained on M^5HisDoc-H works better on MTHv2 than the model trained on MTHv2 and tested on M^5HisDoc-H. The experiments are more interesting if MTHv2 is replaced by another data sets which is not included in M^5HisDoc-H at all.
>
>
> **A3**:  We also conducted cross-validation experiments on other datasets (IC19 HDRC and CASIA-AHCDB). However, due to space constraints, we have included the experimental results in the supplementary material (Section 5.3). The findings of these experiments are consistent with those obtained from MTHv2.
>
>
> > **Q4**: No limitations are discussed in this paper, though some limitations should be, like the one in the above section.
>
>
> **A4**:  We have discussed the limitations of our study in the supplementary material (Section 2). Taking your suggestion into consideration, we will incorporate it into the main text.
>
>
> > **Q5**: Some details are missing, especially the details of the proposed data set.
>
>
> **A5**:  We have included the Datasheets for M^5HisDoc in the supplementary material (Section 1), which encompasses the motivation, composition, collection process, and other pertinent details of M^5HisDoc.

---

> > ### Comment · Reviewer_oBLA · 2023-08-18
> > **resolved some of my concerns**
> >
> > Thanks. The authors' response resolved some of my concerns. I have increased the rating to  "6: Marginally above acceptance threshold".

---

### Official Review · Reviewer_RZNS · 2023-07-24

**Rating:** 6
**Confidence:** 3
**Correctness:** Correct.
**Clarity:** Clear.

**Strengths:**

This dataset is beneficial as a benchmark for ancient texts, and I believe the paper aligns with the intent of the NeurIPS Datasets and Benchmark Track.

The broad application of the baseline method to the fundamental tasks of text-line and character detection/recognition is good. Such baseline results will serve as a foundation for later follow-up studies based on this work.

It is also good that the analysis based on the experimental results is concise. For example, it is stated that rotation and deformation strongly affect the accuracy in L212.

**Additional Feedback:**

None

**Documentation:**

Enough

**Ethics:**

See the limitation

**Limitations:**

My concern is about the license. The authors provided all data by CC BY-NC-ND 4.0. However, the authors denoted that the authors crawled several images from the web (L40). Why can the authors redistribute such images under the CC license? I know copyright does not extend to ancient images (since the authors are not alive). But if an image is from a museum and the museum holds some rights for the image, is it possible to redistribute them under CC?


**Opportunities For Improvement:**

The author could improve Algorithm 1 in the supplemental material (L126). The current description is just a copy-and-paste from Python code.


**Relation To Prior Work:**

Enough

**Summary And Contributions:**

In this paper, the authors construct a dataset of ancient Chinese texts. The created dataset (M5HisDoc) contains more data than other similar datasets (Table 1). In particular, various deformations are considered (Fig. 1). The authors summarize different baseline methods for character detection and recognition and summarize the results.

---

> ### Author Response · Authors · 2023-08-12
> **Response to reviewer RZNS**
>
> Thank you for your valuable time and expertise in the review. Below are our responses to your comments.
>
> ---
>
> > **Q1**: The author could improve Algorithm 1 in the supplemental material (L126). The current description is just a copy-and-paste from Python code.
>
> **A1**:  Thank you for your comment and suggestion. We will improve the description in the final version.
>
>
> > **Q2**: My concern is about the license. The authors provided all data by CC BY-NC-ND 4.0. However, the authors denoted that the authors crawled several images from the web (L40). Why can the authors redistribute such images under the CC license? I know copyright does not extend to ancient images (since the authors are not alive). But if an image is from a museum and the museum holds some rights for the image, is it possible to redistribute them under CC?
>
>
> **A2**:  The images we obtained from the Internet are sourced from some open-copyright websites (such as [Harvard-Yenching Library](https://gj.library.sh.cn/org/harvard), [National Archives of Japan](https://www.digital.archives.go.jp/)). These platforms permit data sharing for non-commercial purposes, therefore, we can make our dataset available under a CC license. Additionally, we will provide the URLs to the raw data.

---

> > ### Comment · Reviewer_RZNS · 2023-08-24
> >
> > Regarding licensing, for the National Archives of Japan, I have confirmed that the materials are freely available; for the Harvard-Yenching Library, I could not find any mention of licensing. Is it mentioned anywhere?
> >
> > In any case, there is no information in the current manuscript about the museums (no description of the National Archives of Japan, for example). I recommend adding the museum name where the authors downloaded images and the policy for redistribution for each museum.

---

> > > ### Author Response · Authors · 2023-08-24
> > > **Response to reviewer RZNS**
> > >
> > > Thank you for your detailed review and constructive suggestions.
> > >
> > > ---
> > >
> > > > **Q1**:  Regarding licensing, for the National Archives of Japan, I have confirmed that the materials are freely available; for the Harvard-Yenching Library, I could not find any mention of licensing. Is it mentioned anywhere?
> > >
> > > **A1**:  We obtained images from the Harvard-Yanjing Library through https://gj.library.sh.cn/org/harvard. It provides download links to the collection of the library. Detailed information about the license (Creative Commons Attribution 4.0 International License) can be found at the bottom-right corner of https://library.harvard.edu/libraries/yenching.
> > >
> > >
> > > > **Q2**:  In any case, there is no information in the current manuscript about the museums (no description of the National Archives of Japan, for example). I recommend adding the museum name where the authors downloaded images and the policy for redistribution for each museum.
> > >
> > >
> > > **A2**:  Thank you for your constructive suggestions. We will add this information in the final version.

---

### Decision · Program_Chairs · 2023-09-22

**Decision:**

Accept (Poster)

**Comment:**

The main contribution of this paper is the M5HisDoc dataset, a complex multi-style Chinese historical document analysis benchmark.

In the first review phase, the reviewers raised some concerns, such as clarity, unclear licensing, somewhat unclear experimental results (e.g., M^5HisDoc-H contains some MTHv2 data), somewhat limited distortions, and a lack of conceptual comparisons between previous datasets.

In the author-reviewer discussion period, all reviewers appreciated the authors' responses and concluded their final review with positive comments.

Considering all the comments by the reviewers, I am recommending acceptance.